# Structural Evidence of Spin State Selection and Spin Crossover Behavior of Tripodal Schiff Base Complexes of tris(2-aminoethyl)amine and Related Tripodal Amines

**Greg Brewer**

Department of Chemistry, Catholic University of America, Washington, DC 20064, USA; brewer@cua.edu

**Abstract:** A review of the tripodal Schiff base (SB) complexes of tris(2-aminoethyl)amine, $N_{ap}(CH_2CH_2NH_2)_3$ (tren), and a few closely related tripodal amines with Cr(II), Mn(III) ($d^4$), Mn(II), Fe(III) ($d^5$), Fe(II) ($d^6$), and Co(II) ($d^7$) is provided. Attention is focused on examination of key structural features, the $M-N_{imine}$, $M-N_{amine}$, or M-O and $M-N_{ap}$ bond distances and $N_{imine}$-M-N(O) bite and $C-N_{ap}$-C angles and how these values correlate with spin state selection and spin crossover (SCO) behavior. A comparison of these experimental values with density functional theory calculated values is also given. The greatest number, 132, of complexes is observed with cationic mononuclear iron(II) in a $N_6$ donor set, $Fe(II)N_6$. The dominance of two spin states, high spin (HS) and low spin (LS), in these systems is indicated by the bimodal distribution of histogram plots of $Fe(II)-N_{imine}$ and $Fe(II)-N_{azole/pyridine}$ bond distances and of $N_{imine}–Fe(II)-N_{azole/pyridine}$ and $C-N_{ap}$-C bond angles. The values of the two maxima, corresponding to LS and HS states, in each of these histograms agree closely with the theoretical values. The iron(II)-$N_{imine}$ and iron(II)-$N_{azole/pyridine}$ bond distances correlate well for these complexes. Examples of SCO complexes of this type are tabulated and a few of the 20 examples are discussed that exhibit interesting features. There are only a few mononuclear iron(III) cationic complexes and one is SCO. In addition, a significant number of supramolecular complexes of these ligands that exhibit SCO, intervalence, and chiral recognition are discussed. A summary is made regarding the current state of this area of research and possible new avenues to explore based on analysis of the present data.

**Keywords:** spin crossover; tripodal ligands; supramolecular; structure; transition metal complex; hydrogen bonding; imidazole and pyrazole

## 1. Introduction

<u>Spin crossover complexes.</u> Spin crossover complexes have been observed for octahedral complexes of $d^4$–$d^7$ metals of the first transition series [1]. This unique feature occurs when the crystal field stabilization energy (CFSE), $\Delta_{Oh}$, is equal to the pairing energy (PE). The PE is the energy required to pair the largest possible number of the d electrons in the $t_{2g}$ orbital set to produce a ground state with the lowest possible multiplicity. The multiplicity is 2S+1 where S is the spin quantum number of the metal. A complex in which $\Delta_{Oh}$ = PE has two electronic ground states, high spin (HS) and low spin (LS), which differ only slightly in energy at the crossover point. SCO complexes have a reversible change in the ground state, which can be affected by a slight change in temperature, pressure, and absorption of appropriate electromagnetic radiation [2]. The change is easily followed by variable temperature magnetic or structural studies as well as other physical/spectroscopic techniques [3].

The foci of this review, as given in the title, are the structural features that indicate a preference for one spin state over another and structural feature changes that accompany a SCO in tripodal Schiff bases of tris(3-aminoethyl)amine, tren. This approach is based on a comment from Nobel Laureate R. Hoffmann. "There is no more basic enterprise in chemistry than the determination of the geometrical structure of a molecule. Such a determination, when it is well done, ends all speculation as to the structure and provides us with the starting point for the understanding of every physical, chemical and biological property of the molecule". Structure determines the magnetic properties. Magnetic properties do not determine the structure. The structural approach to investigating SCO complexes has been followed in earlier SCO reviews [4–6].

The SCO may be described as abrupt or gradual, complete or incomplete and may occur with or without hysteresis. $T_{1/2}$ is used to describe the position of SCO and is the temperature at which the HS and LS populations are present at 50:50 levels. The HS state has longer M-ligand bond distances, as well as other structural differences, than the LS state. The magnitude of the bond distance change varies from 0.2 to 0.1 Å depending on the metal involved. A SCO involving a $d^6$ or a $d^5$ metal, for example, LS iron(II) or cobalt(III) ($t_{2g}^6 e_g^0$) to HS iron(II) or cobalt(III)($t_{2g}^4 e_g^2$) or LS iron(III) or manganese(II) ($t_{2g}^5 e_g^0$) to HS iron(III) or manganese(II) ($t_{2g}^3 e_g^2$), involves movement of two electrons and thus a large change in bond distance, 0.2 Å. Alternately a SCO involving a $d^4$ or a $d^7$ metal, for example, LS manganese(III) or chromium(II) ($t_{2g}^4 e_g^0$) to HS manganese(III) or chromium(II) ($t_{2g}^3 e_g^1$) or LS cobalt(II) ($t_{2g}^6 e_g^1$) to HS cobalt(II) ($t_{2g}^5 e_g^2$), involves movement of one electron and thus a smaller change in bond distance, 0.1 Å. For the $d^5$ and $d^6$ systems it is possible to have a two step transition between the LS and HS states by going through the intermediate spin (IS) state [7], $t_{2g}^4 e_g^1$ for $d^5$ or $t_{2g}^5 e_g^1$ for $d^6$. SCO complexes exhibit a bistability and are certainly important as they have applications ranging from the iron(III) enzyme cytochrome P450 [8] to materials research including switches [9,10] and memory storage [11]. This is because they can exist in either of two electronic states under slightly different controllable environmental or chemical conditions, such as pH, hydrates/solvation, hydrogen bonding, and presence of guests in the lattice of SCO complex [12,13].

<u>Tripodal ligands</u>. A great variety of ligands complexed to appropriate metals have been shown to exhibit SCO behavior (Figure S1). The tripodal ligands with a tren (or closely related) backbone are the focus of this review. They have features that are easy to exploit in the search for new SCO complexes. An indication of the importance of tren to SCO is that a tren ligand provided the first case of a SCO complex of $d^4$ manganese(III) [14]. Tren can undergo simple Schiff base condensations with a variety of aldehydes to generate ligands that have varying donor sets. The donor set is a listing of the atoms that are directly bound to the metal. The tren of the octahedral tren Schiff bases (SB) provides three imine nitrogen donor atoms (those at the end of the three arms). The aldehyde provides the remaining three donor atoms, either three oxygen or nitrogen atoms. The donor sets provided by this class of ligands to date are the $N_6$, $N_3O_3$ (both structurally characterized), and $N_3N_aO_b$ (a + b = 3) set (synthesized but not structurally characterized to date). This variation in donor set allows for tuning of the complex to match the SCO condition, $\Delta_{Oh}$ = PE. There are structural variations of the tren backbone that generate an even greater variety of ligands.

The tripodal Schiff base ligands are made up of two components, the tripodal amine (backbone) and the aldehyde that condenses to it. Examples of the tripodal amines and aldehydes that they react with are shown in Figure 1. Line drawings of two of the fully formed ligands are shown in Figure 1 as well. A Supplementary Material file gives line drawings of all the ligands and their abbreviations that are discussed in the review. There are three choices regarding nomenclature of the ligands. These are to (1) label each as $L^1$ and $L^2$ and so on, (2) provide the IUPAC name of the ligand or (3) provide an abbreviation for the ligand. The option followed here is the last, as this is descriptive of the two components of the ligand (backbone and aldehyde) and has been used by several researchers in the field. The first method provides no descriptive information of the ligand and the IUPAC name is exceptionally cumbersome to use. Compare the IUPAC name (tris(3-aza-4-(6-methyl-2-pyridyl)but-3-eneyl)amine) to the abbreviation employed here, tren(6Me-Py)$_3$.

$N(CH_2CH_2NH_2)_3$

tren

$N(CH_2CH_2CH_2NH_2)_3$

trpn

$CH_3\overset{\oplus}{N}(CH_2CH_2NH_2)_3$

N-Metren

$HC(CH_2CH_2NH_2)_3$

tram

2ImH

4ImH

Pyr

PyrzH

4Thia

Sal

Py

N-oxoPy

2Thia

$N\text{-}(CH_2CH_2N{=}CH—\text{...})_3$

tren(1Me-2Im)$_3$

$N\text{-}(CH_2CH_2N{=}CH—\text{...})_3$

tren(5Me-4ImH)$_3$

**Figure 1.** Line drawings of tripodal amines (top), aldehydes (middle), and examples of tripodal Schiff base ligands (bottom) with names. A particular tripodal ligand is specified by giving the symbol of the amine, which serves as the backbone, followed by the aldehyde that it reacts with. A line drawing of each of the ligands utilized and its abbreviation are provided as a supplemental file.

Structural survey of the Cambridge Structural Database (CSD). The CSD was used to find structurally characterized examples that met the search criteria. The search criteria are specified in Table 1 and indicate the Schiff base backbone, presence, and type of metal, whether metal is bound to the three Schiff base imine nitrogen atoms, and whether the metal is bound to the three Schiff base imine nitrogen atoms plus other donors (3 oxygen or 3 nitrogen atoms). The number of hits simply indicates the literature activity in any category. Examples of the mononuclear complexes of first row $d^4$–$d^7$ metals and a correlation of their structural parameters with spin state selection will be discussed in subsequent sections. The temperature variation of the structural parameters of SCO complexes will also be given.

**Table 1.** Survey of the number of structurally characterized tren Schiff base compounds. The central or apical nitrogen atom of the tripodal amine is labeled to as $N_{ap}$.

| System | Number of Compounds | System | Number of Compounds |
|---|---|---|---|
| $N_{ap}(CH_2CH_2N{=}C)_3$ | 890 | $C{-}N_{ap}{}^+(CH_2CH_2N{=}C)_3$ | 3 |
| $N_{ap}(CH_2CH_2N{=}C)_3$ + any metal | 764 | $HC(CH_2CH_2N{=}C)_3$ | 2 |
| $N_{ap}(CH_2CH_2N{=}C)_3$ + any transition metal | 563 | $N_{ap}(CH_2CH_2CH_2N{=}C)_3$ | 19 |
| $N_{ap}(CH_2CH_2N{=}C)_3$ + any transition metal bound to 3 Schiff base N atoms | 517 | $N_{ap}(CH_2CH_2N{=}C)_3$ + any transition metal bound to 3 Schiff base N atoms + 3 mixed N, O, or 3 S atoms | none |
| $N_{ap}(CH_2CH_2N{=}C)_3$ + any transition metal bound to 3 Schiff base N atoms + 3 other non-metal atoms | 431 | $Nap(CH_2CH_2N{=}C)_3$ + any transition metal bound to 3 Schiff base N atoms + 3 other O | 92 |
| $N_{ap}(CH_2CH_2N{=}C)_3$ + any transition metal bound to 3 Schiff base N atoms + 3 other N atoms | 339 | $N_{ap}(CH_2CH_2NH{-}CH)_3$ + any transition metal bound to 3 reduced Schiff base N atoms + 3 other non metals | 19 |

Polynuclear systems will also be discussed. Table 2 provides background numbers to indicate the number of tren systems that are in the database that illustrate the extent of work in this area. The results of this broad structural survey show that all tren SB transition metal complexes have a $N_6$ (339) or $N_3O_3$ (92) ligand donor set, which may indicate a homoleptic preference. To date, the only complexes of asymmetric ligands that have been characterized differ only in protonation state, for example, $tren(2ImH)_2(2Im)^{1-}$ vs. $tren(2ImH)_3$. The few reported asymmetric complexes such as $Fetren(sal)_2(pyr)$ and $Fetren(sal)(pyr)_2$ [15], which have $N_4O_2$ and $N_5O$ donor sets, respectively, have not been structurally characterized.

## 2. Structural Considerations

<u>Ligand binding</u>. The Schiff base ligands discussed here bind to the metal to give an approximate octahedron. The three imine nitrogen atoms of the tren bind facially as do the set of other three donor atoms, N or O, provided by the aldehyde. The central nitrogen atom of the tripodal amine, $N_{ap}$, caps the trigonal face formed by the three imine donor atoms. Schiff base ligands formed from imidazole or pyrazole carboxaldehyde are triprotic, $H_3L$, even when bound to a metal. These ligands can bind as $H_3L$, $H_2L^-$, $HL^{2-}$, or $L^{3-}$. In addition they can bind to a metal in the hemideprotonated form, $H_{1.5}L^{1.5-}$. The hemideprotonated ligand can be thought of as a 50/50 mixture of fully protonated and fully deprotonated ligand; or in some instances it is actually $H_{1.5}L^{1.5-}$ due to a crystallographically imposed hydrogen atom occupancy of 0.5. An example of this diversity in binding is given by $tren(4ImH)_3$. This ligand can bind to iron(III) in the fully protonated form to give $[Fetren(4ImH)_3]^{3+}$ in the fully deprotonated form to give $Fetren(4Im)_3$, or in the hemideprotonated form to give $[Fetren(4ImH_{0.5})_3]^{1.5+}$. Data on these complexes as well as related ones will be given in subsequent sections. A significant advantage of the triprotic ligands is that the field strength of the ligand is increased on deprotonation $\Delta_{Oh}(H_3L) < \Delta_{Oh}(H_2L^-) < \Delta_{Oh}(H_{1.5}L^{1.5-}) < \Delta_{Oh}(HL^{2-}) < \Delta_{Oh}(L^{3-})$, which allows for greater control on reaching the SCO condition. In addition, the triprotic ligands can form self-assembled dimeric, tetrameric, or extended 2D supramolecular systems in which the important properties of SCO, chiral recognition, intervalence, and magnetic exchange (ferromagnetic and antiferromagnetic) can all be present in the same molecular assembly.

<u>Tripodal amine backbone conformation</u>. The central apical nitrogen, $N_{ap}$, atom is nominally $sp^3$ hybridized and two conformations of the tripodal amine were originally reported [16]. These two conformations are planar, "P", where the bond angles around $N_{ap}$ are 120°, and "$N_{ap}$ in" where $N_{ap}$ is bent in slightly toward the metal with a corresponding reduction in bond angle.

"$N_{ap}$ in" and "P" are associated with the HS and LS state respectively. The two conformations seen in a SCO crossover complex are the "P" and "$N_{ap}$ in".

$$"P" \text{ (LS)} \rightleftarrows "N_{ap} \text{ in}" \text{ (HS)}.$$

A third conformation is "$N_{ap}$ out" where $N_{ap}$ is approximately sp$^3$ hybridized and bent away from the metal. To   date it has only been observed with SB complexes of N-Metren [17] or tram [18] (see Figure 1). The positive charge on the quaternized $N_{ap}$ atom of N-Metren is repelled by the metal cation of the complex resulting in an extreme $N_{ap}$–M non bonded distance of 3.9 Å. In tama, the apical atom is a CH; but as the carbon atom has four substituents, it adopts the same geometry as do those of N-Metren with a very long C-M non bonded distance, 3.8 Å. The N-Metren and tama SB complexes appear to be conformationally locked into a LS state. A fourth conformation is an "extreme $N_{ap}$ in" (capped trigonal antiprism, coordination number of seven, CN = 7). In this case $N_{ap}$ is bent in so close to the metal that it must be regarded as a seven coordinate complex. The "extreme $N_{ap}$ in" conformation is always observed with SB complexes of trpn [19]. The extra carbon atom of the trpn complexes provides enough flexibility to allow the $N_{ap}$ atom to form a long (2.5 Å) seventh bond to the metal. Ironically the formation of the seventh bond forces a HS state as it comes at the expense of lengthening the other six metal bond lengths. It is also observed with reduced $N_{ap}(CH_2CH_2NHCH)_3$ complexes of tren [20].

There is a clear correlation of the conformation of the backbone with spin state selection. The $N_{ap}$–M distance varies in a continuous manner and the terms "$N_{ap}$ out", "P", "$N_{ap}$ in", and extreme "$N_{ap}$ in" are simply descriptive of points on this scale. Each of these conformations is illustrated with iron(II) SB complexes in Figures 2 and 3. The three arms of the tripodal ligand can wrap around the metal in a clockwise (Δ) or counterclockwise (Λ) manner. Care was taken to show all the complexes with the same twist. The diagrams of this review are all prepared by the author from the original cif but are not given as ORTEPS. This allows for a clearer and consistent representation of all of the molecules depicted to cleanly illustrate the desired points. This method allows for easy elimination of solvents and anion (often disordered) and for overlay of structures when necessary to clearly illustrate structural changes that occur on SCO. This feature is frequently utilized by researchers in the area when not simply presenting a new structure.

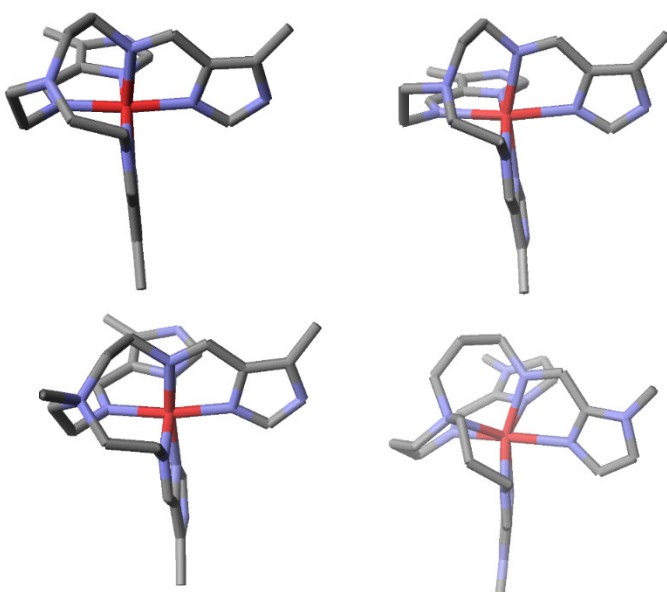

**Figure 2.** Structural diagrams of four iron(II) complexes illustrating the four backbone geometries described above. Hydrogen atoms and anions have been omitted for clarity. Nitrogen atoms are in blue and iron in red. The complexes from top left to bottom right are low spin (LS) [Fetren(5Me-4ImH)$_3$ ]$^{2+}$(P) [17], high spin (HS) [Fetren(5Me-4ImH)$_3$ ]$^{2+}$ ($N_{ap}$ in") [17], LS [FeNMetren(5Me-4ImH)$_3$ ]$^{3+}$ [17] ($N_{ap}$ out), and HS [Fetrpn(1Me-2Im)$_3$]$^{2+}$ ("extreme $N_{ap}$ in") [19]. Please note that [FeNMetren(5Me-4ImH)$_3$ ]$^{3+}$ is an iron(II) complex even though its charge is 3+, which is due to the quaternized $N_{ap}$.

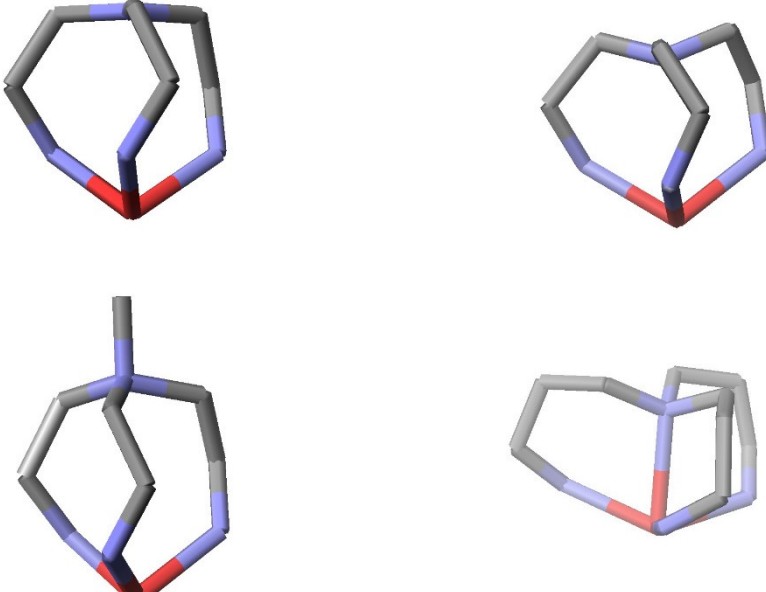

**Figure 3.** Structural diagrams of the four iron(II) complexes of Figure 2 showing only the iron and backbone atoms of tripodal ligands. The orientations place $N_{ap}$ directly above the iron atom. The structural parameters, Fe-$N_{imine}$, Fe-$N_{axole}$, Fe-$N_{ap}$, $N_{imine}$–Fe-$N_{azole}$, and C-$N_{ap}$–C, of these complexes are as follows: LS [Fetren(5Me-4ImH)$_3$ ]$^{2+}$(P) (1.978 Å, 1.972 Å, 3.527 Å, 80.9970°, 119.829°) [17], HS [Fetren(5Me-4ImH)$_3$]$^{2+}$ ($N_{ap}$ in") (2.185 Å, 2.206 Å, 3.050 Å, 75.665°, 115.454°) [17], LS [FeNMetren(5Me-4ImH)$_3$ ]$^{3+}$ ('$N_{ap}$ out) (1.956 Å, 1.960 Å, 3.920 Å, 81.867°, 113.554°) [17], and HS [Fetrpn(1Me-2Im)$_3$]$^{2+}$ ("extreme $N_{ap}$ in"). (2.2746 Å, 2.2437 Å, 2.4476 Å, 72.8°, 107.79°) [19].

It should be noted that the experimental values for the LS and HS iron(II) SCO complex, [Fetren(5Me-4ImH)$_3$ ]$^{2+}$ [17], given above are consistent with density functional theory, DFT, theoretical values from Table 2, which are provided in the next section.

[Fetren(5Me-4ImH)$_3$ ]$^{2+}$ "P" (LS) $\rightleftarrows$ [Fetren(5Me-4ImH)$_3$ ]$^{2+}$ "$N_{ap}$ in" (HS).

$^1$A $\rightleftarrows$ $^5$E

<u>DFT structural parameters of HS and LS</u>. Earlier work involving density functional theory calculations was done on LS and HS iron(II), $d^6$, and iron(III), $d^5$, tripodal imidazole complexes of tren [21,22]. A few key structural results that are predictors of spin state are given in Table 2. It was found that the parameters depend on the spin state much more than the oxidation state. In other words, the change in the spin state is more important (structurally) than a change in the oxidation state. These theoretical structural parameters will be compared to specific complexes taken from the experimental CSD in subsequent sections. It should be pointed out that direct comparison of these values to structural values from the literature should be most valid for iron complexes of five membered cyclic aromatic aldehydes, imidazole, pyrazole, and pyrolle carboxaldehydes. Comparison to complexes of other metals or other aldehydes such as the six membered ring examples of salicylaldehyde, 2-pyridine carboxaldehyde, and pyridine-2-carboxaldehyde N-oxide can be made qualitatively, for example, to predict the lengthening or shortening of a distance. There will be anticipated differences between the values in Table 2 and structural data for a 6 membered ring aldehyde rather than a 5 membered ring aldehyde. The bite angle should widen for the six membered ring for steric reasons and the M-$N_{(Imidazole, pyrazole, pyrrole)}$ parameter now must be replaced with a M-$N_{pyridine}$ or a M-O distance.

**Table 2.** Theoretical structural parameters [21–22] for LS and HS complexes of iron(II) or iron(III) tren(imidazole)₃ complex.

| Structural Parameter | Low Spin | High Spin |
|---|---|---|
| M-N$_{imine}$ (Å) | <2.00 Å | >2.10 Å |
| M-N$_{(Imidazole, pyrazole, pyrrole)}$ (Å) | <2.00 Å | >2.10 Å |
| M-N$_{ap}$ (Å) | 3.50 Å | 3.00 Å |
| N$_{imine}$–M–N bite angle° | 81° | 76° |
| C-N$_{ap}$-C ° | "P" 120° | "N$_{ap}$ in" 114° |

## 3. Mononuclear Metal Systems

<u>Chromium(II) d⁴</u>. The first SCO chromium(II) complexes were Cr(depe)I₂ (depe = 1,2-bis(ethylphosphino)ethane [23,24] followed by the sandwich compound [η⁵ Cp*-Cr – η⁵ P₅ –Cr- η⁵ Cp*]X (X = PF₆⁻ or SbF₆⁻) [25]. These organometallic ligands are quite different from the Schiff bases of this discussion. Schiff base complexes of chromium(II) are rather rare due to the experimental difficulty of working with the oxidatively unstable chromium(II) complexes. In addition, the strong field ligands, necessary to stabilize the LS state and therefore SCO complexes, only increase their ease of aerial oxidation, which require dry box or related inert atmosphere techniques. There is only one structural report of a chromium(II) complex of a tripodal Schiff base condensate. The ligand is the condensate of tren with 5-methycarboxolyate-2-pyridinecarboxaldehyde [26]. It is reported as a LS chromium(II) complex, [CrL]²⁺, with a magnetic moment, μ, Bohr magnetons(BM) of 3.40 BM at 298 and 2.83BM at 5 K. The structure at 120 K is shown in Figure 4. The magnetic moment values and structural parameters suggest the possibility of a LS Cr(II) complex or even an incomplete SCO species. However the authors report that the magnetic data is also explained by an internal electron transfer from Cr(II) to L to give a [Cr(III) L·⁻]²⁺ radical anion species. The temperature dependence of magnetic moment can be explained by antiferromagnetic coupling of the chromium(III) with the radical anion. The critical structural parameters of the [Cr(III)L]³⁺ complex [27], prepared independently, are essentially identical to the [Cr(II)L]²⁺ values. See Figure 4.

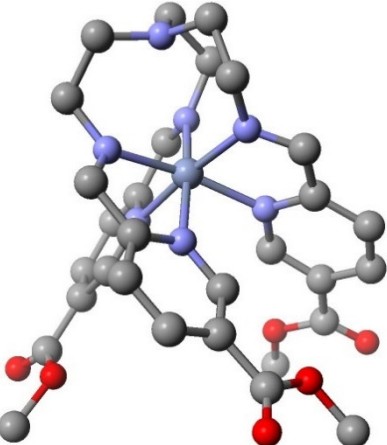

**Figure 4.** Structure of a tripodal SB chromium(II) complex. The hydrogen atoms have been removed for clarity. The structural parameters, Cr-N$_{imine}$, Cr-N$_{pyridine}$, Cr-N$_{ap}$, N$_{imine}$ –Cr-N$_{pyridine}$, and C-N$_{ap}$ –C, for the Cr(II) and Cr(III) complexes are as follows, Cr(II) 1.989 Å, 2.027 Å, 3.086 Å, 79.040°, and 117.685° and Cr(III) 2.050 Å, 2.067 Å, 3.120 Å 79.315°, and 116.671°.

The above N₆ ligand is capable of producing strong fields, as demonstrated by the analogous [FeL]²⁺ complex, which is LS [26]. However, the evidence for the chromium complex can be interpreted as a SCO chromium(II) or a chromium(III) complex of a ligand radical anion. The ease of oxidation of chromium(II) supports the latter.

The only other structure of a Cr(II) SB with tren is a 5-coordinate compound in which the three imine nitrogen atoms, the apical nitrogen atom and a chloride ion are bound to a chromium(II) [28]. <u>Manganese(II), $d^5$, and manganese(III), $d^4$</u>. There are 48 reported structures of a manganese atom bound to the three imine nitrogen atoms of a tren tripodal SB and three other non metal atoms, 23 to three nitrogen atoms, and 25 to three oxygen atoms. All cationic species are of a manganese(II) and all neutral species are of manganese(III) regardless of donor set, $N_6$ or $N_3O_3$.

The most important example is of Mntren(Pyr)$_3$, which is the only known SCO complex of manganese(III) and the first $d^4$ SCO complex. Fetren(Pyr)$_3$ (prepared before the manganese complex) is LS [29] at all temperatures. This prompted the synthesis and investigation of Mntren(Pyr)$_3$ since $\Delta_{Mn(III)}$ is greater than $\Delta_{Fe(III)}$. Mntren(Pyr)$_3$ exhibits a rather sharp SCO at 60 K.

$$\text{Mn(III) } (t_{2g}{}^4 e_g{}^0) \ (^3T, LS, S = 1) \rightleftarrows \ \text{Mn(III) } (t_{2g}{}^3 e_g{}^1) \ (^5E, HS, S = 2)$$

The structures have been determined at temperatures below and above the SCO point [30]. Figure 5 shows the superimposed structures of Mntren(Pyr)$_3$ at 30 K and 295 K. Note that the donor atoms of the green LS form are closer to the metal than those of the yellow HS form. The structural parameters of the LS and HS forms of Mntren(Pyr)$_3$ are Mn-N$_{imine}$ distance (2.027Å, 2.124 Å), Mn-N$_{pyrole}$ distance (1.975 Å, 2.048 Å), Mn-N$_{ap}$ distance (3.284 Å, 3.233 Å), N$_{imine}$–Mn-N$_{pyrolee}$ angle (80.399°, 78.350°), and the C-N$_{ap}$–C angle (118.833°, 117.415°). All of the experimental parameters are consistent with the theoretical predictions from Table 2. While undergoing the SCO (LS to HS), all bond distances lengthened, the angles narrowed, and the non bonded Mn-N$_{ap}$ distance decreased. The structural changes with Mn(III) ($d^4$) were much smaller than those of iron(II) ($d^6$) as the SCO involves only a one vs. a two electron change. There are no other known LS complexes of Mn(III) other than those of cyanide and therefore no SCO complexes. The other manganese(III) tren complexes (22 in total) are of the condensates of tren with substituted salicylaldehydes or pyridine N-oxide-2-carboxaldehyde, which afford the $N_3O_3$ donor set. There were no examples of LS or SCO with Mn(II) and the other tripodal $N_6$ ligands. The experimental values of the key structural parameters of these manganese(II) and manganese(III) species were all consistent with the experimentally observed HS assignment.

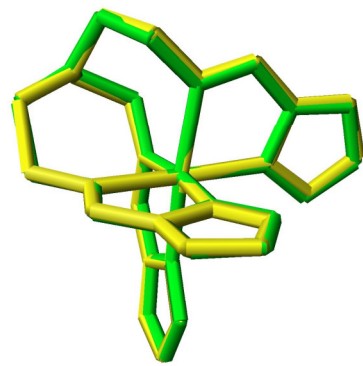

**Figure 5.** Superimposed structures of LS (green, 30 K) and HS (yellow, 293 K) Mntren(Pyr)$_3$.

It is disappointing that to date there are no SCO complexes of Mn(II) to accompany that found for Mn(III). Recently the synthesis and structure of the supramolecular Mn(II) system, {[Mntren(4ImH$_{0.5}$)$_3$]$_2$[Mntren(4ImH)$_3$]$_2$}(ClO$_4$)$_5$}, was reported [31]. The authors have succeeded in creating a hemideprotonated Mn(II)N$_6$ species as part of the complex. The ligand field strength of H$_3$L increases on deprotonation of the triprotic imidazole ligands, H$_3$L. However, the structural parameters of the hemideprotonated manganese(II) in the above complex are still consistent with a HS assignment.

Iron complexes. As with other families of SCO ligands the field is dominated by iron(II) and iron(III). There are 209 complexes containing an iron atom bound to the three imine nitrogen atoms of a tripodal tren Schiff base ligand. This number is not overwhelming and not all of these are SCO but a good organizational scheme is needed to understand these systems. The mononuclear complexes will be discussed first and polynuclear and supramolecular systems will be discussed at the end of this report. Within the mononuclear complexes several divisions will be made, cationic iron(II) complexes (132), which include those of a five membered ring containing a carboxaldehyde (83) and those of a six membered ring containing a carboxaldehyde (49). There are no mononuclear neutral or anionic iron(II) complexes. There are iron(III) cationic complexes (6) and iron(III) neutral complexes (15).

Iron(II) cationic complexes with a 5 membered ring carboxaldehyde. The $N_6$ donor set has produced a wide variety of SCO complexes with iron(II) [32] and the tripodal Schiff bases exhibit this same behavior. An aspect of the cationic iron(II) tren $N_6$ complexes that should be mentioned is their redox stability relative to the analogous iron(III) complexes. Attempts to prepare Fetren(Py)$_3^{3+}$ or Fetren(N-Me-2-Im)$_3^{3+}$ by direct reaction of the preformed ligand in air with iron(III) salts have resulted in high yields of the reduced Fetren(Py)$_3^{2+}$ and Fetren(1Me-2Im)$_3^{2+}$ salts [33]. These salts were structurally characterized and found to be identical to samples of the iron(II) complexes made by reaction of the ligand with iron(II) salts. The implications of this unusual reduction in air of iron(III) $N_6$ complexes to the analogous iron(II) $N_6$ complexes are not clear. However, it is an aspect of the chemistry of these species that is not fully understood and when studied may result in the ability to produce more iron(III) $N_6$ cationic complexes such as Fetren(Py)$_3^{3+}$ or Fetren(N-Me-2-Im)$_3^{3+}$ that are presently not known.

In SCO complexes the dominant structural change is an increase in the M-Donor bond distance with increasing temperature. For iron(II) this is about 0.2 Å. In the absence of a SCO the M-Donor distance for a series of complexes will vary considerably. The range can be tightened if the oxidation state of the metal, donor set of the ligands, coordination number, and identity of the donor are all held constant. The M-Donor distance range can be narrowed further if the ligand is hexadentate, which places even greater constraint on bond distances. The Fe(II) Schiff base complexes of tren condensed to a five membered ring aldehyde containing a donor nitrogen atom meet all these criteria. In the absence of a SCO there would still be variation of the Fe-N$_{(imine\ or\ azole)}$ bond distance but that variation should be rather small and evenly displaced about the average. The same would be true of the other structural parameters from Table 2. Histograms of selected structural parameters from Table 2 are pictured in Figures 6 and 7 and clearly show a bimodal distribution. The interpretation of this bimodal distribution is that these complexes have two different electronic ground states since a single ground state (which has a fixed structure) should produce a single maximum. For these iron(II) $N_6$ cationic complexes, the bimodal distribution is taken as evidence of the HS and LS states. Further support of this comes from comparison of the theoretical Fe-N bond distances from Table 2 with the analogous values from the histograms of Figure 6. Both give Fe-N bond distances of 1.97(LS) and 2.20 Å (HS). Extension of this idea is shown in the histograms of the N$_{imine}$ –Fe(II)-N$_{azole}$ bite and C-N$_{ap}$-C angles shown in Figure 7. The LS and HS bite angles from Figure 7 of 75° and 81°, for HS and LS complexes, respectively, are very close to the values from Table 2. Similarly, the C-N$_{ap}$-C maximum of 114° for HS complexes and 120° for LS complexes are in good agreement with the values from Table 2.

Figure 8 is a scatterplot of the Fe-N$_{imine}$ and Fe-N$_{azole}$ bond distances, which has a correlation of $R^2$ = 0.806. The general correlation of these parameters is due to the fact that both donor atoms are from the same arm of the ligand. These two atoms cannot act totally independently as they are tethered together by the backbone of the ligand and the fact that both bind to the iron atom. They are in lockstep with one another. Figure 8 also shows a bimodal cluster of points, those at the lower left are LS complexes (both short Fe-N$_{imine}$ and Fe-N$_{azole}$ distances) and those at the upper right are HS complexes (both long Fe-N$_{imine}$ and Fe-N$_{azole}$ distances). This is due to the lockstep relationship of these parameters. There is a small grouping of points well above the line that have short Fe-N$_{imine}$ and long Fe-N$_{azole}$ distances. These compounds are of [Fetren(Thia)$_3$]$^{2+}$ (tren condensed with

thiazolecarboxaldehyde). The presence of the sulfur atom in the ring must distort the electron distribution enough to lengthen the Fe-N(5-membered ring) bond.

There was not a correlation of the bite angle with SCO but there was a correlation of the bite angle with the spin state. Bite angles of 76° favored a HS state while LS was favored for bite angles of 81°. The compression or widening of this angle was due to the Fe-N bond distance change that occurs with the SCO. Shortening of both Fe-N bonds (they must act in concert) caused a widening of the angle while lengthening caused the opposite effect.

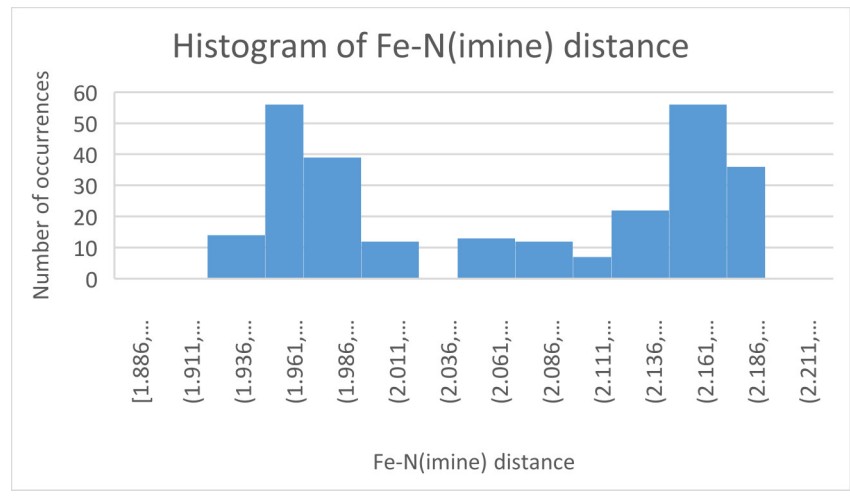

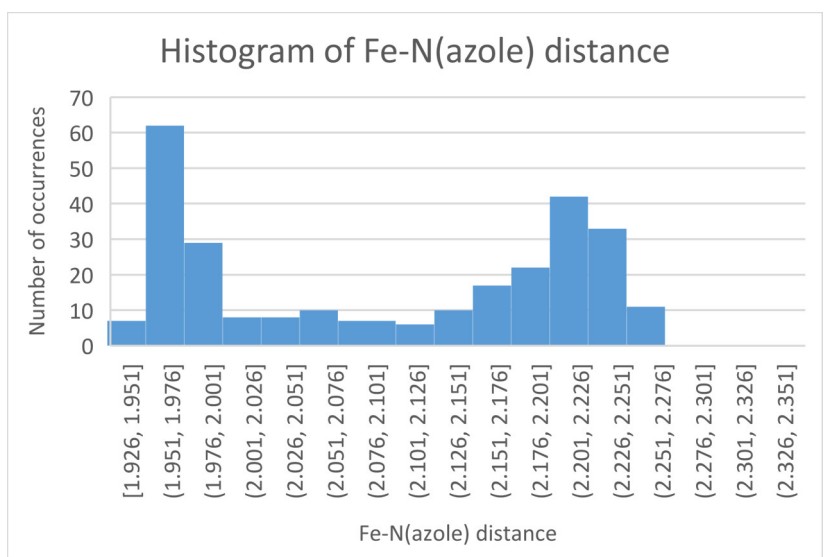

**Figure 6.** Histogram of the Fe(II)-$N_{imine}$ bond distance (top) and Fe(II)-$N_{azole}$ bond distance (bottom) of the 83 cationic compounds of a Schiff base (SB) of tren condensed with a 5 membered azole carboxaldehyde. Note the bimodal distribution.

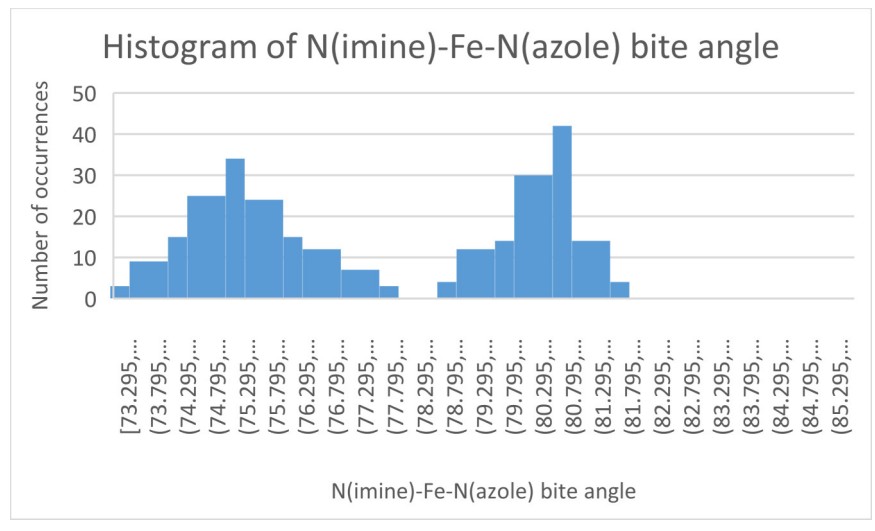

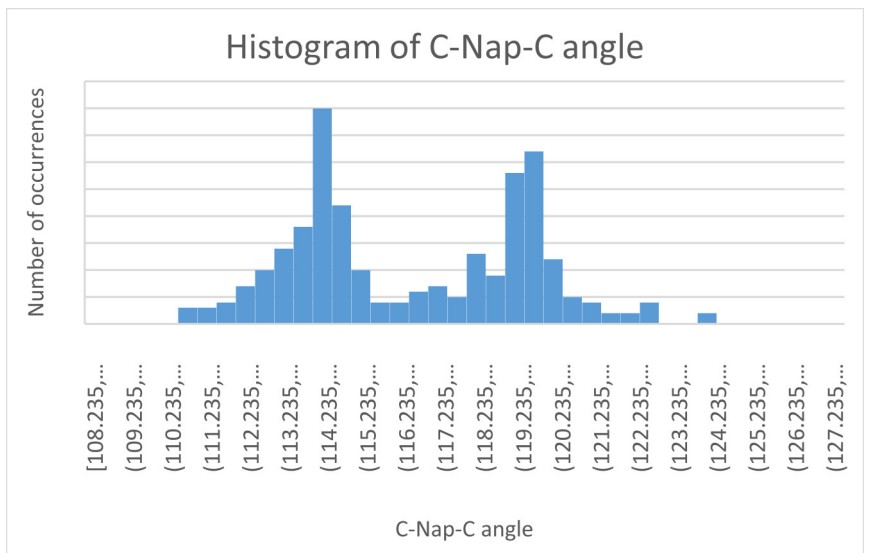

**Figure 7.** Histogram of the $N_{imine}$-Fe(II)-$N_{azole}$ bite angle (top) and C-$N_{ap}$-C bond angle (bottom) of 83 cationic compounds of a SB of tren condensed with a 5 membered azole carboxaldehyde. Note the bimodal distribution.

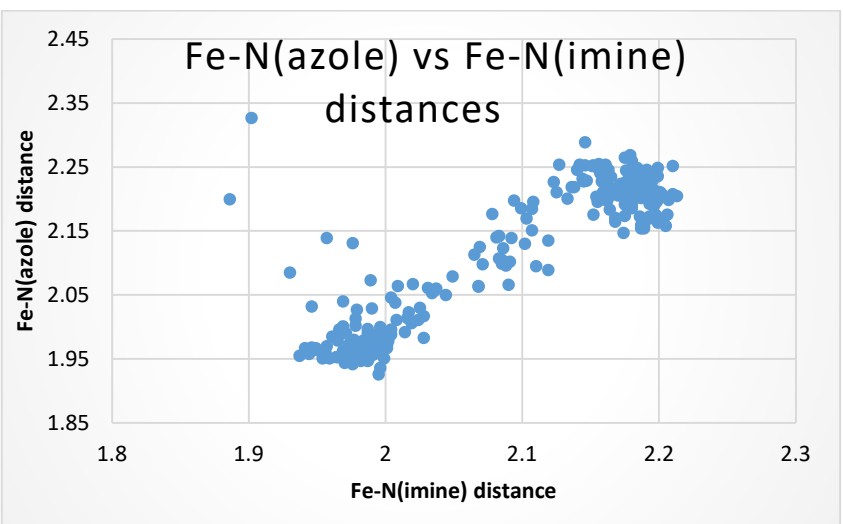

**Figure 8.** Scatterplot of the Fe-$N_{imine}$ (x axis) vs. Fe-$N_{azole}$ (y axis) bond distances of the 83 cationic compounds of a SB of tren condensed with a 5 membered azole carboxaldehyde. $R^2 = 0.806$.

Data for the LS and HS forms of SCO complexes of cationic mononuclear iron(II) $N_6$ complexes are given in Table 3. More of the 83 complexes of this class may be SCO but the ones in Table 3 were selected since there was structural data available for both spin states.

**Table 3.** Structural parameters, Fe-$N_{imine}$, Fe-$N_{5\ membered\ ring}$, Fe-$N_{ap}$ (Å) $N_{imine}$ –Fe-$N_{5\ membered\ ring}$ bite, and C-$N_{ap}$ –C angles (°) of LS and HS iron(II) SCO complexes with Schiff base condensates of tren with five membered ring aldehydes as ligands.

| NAME/CSD file | T | Fe-$N_{imine}$ | Fe-$N_{azole}$ | Fe-Nap | $N_{imine}$-Fe-$N_{azole}$ | C-$N_{ap}$-C |
|---|---|---|---|---|---|---|
| Fe[tren(4ImH)₃](BF4)₂ | | | | | | |
| EBUQAU | 108 | 1.989 | 1.981 | 3.522 | 81.084 | 119.955 |
| EBUQAU0 | 293 | 2.196 | 2.189 | 2.888 | 75.776 | 113.91 |
| Fe[tren(2ImH)₃](BF4)₂ | | | | | | |
| ENUBAR | 100 | 1.985 | 1.961 | 3.473 | 81.163 | 119.494 |
| ENUBAR02 | 295 | 2.175 | 2.174 | 2.976 | 76.218 | 114.781 |
| Fe [tren(1-nBu-2Im)₃](PF6)₂ | | | | | | |
| GIGGUA07 | 25 | 1.986 | 1.978 | 3.423 | 81.002 | 119.265 |
| GIGGUA09 | 230 | 2.174 | 2.225 | 3.001 | 74.753 | 113.68 |
| Fe[tren(PyrzH)₃](NO3)₂ | | | | | | |
| JIQDET05 | 30 | 2.017 | 2.023 | 3.413 | 79.288 | 118.934 |
| JIQDET | 300 | 2.184 | 2.249 | 2.96 | 75.025 | 114.401 |
| Fe[tren(1-nBu-2Im)₃](ClO4)₂ | | | | | | |
| JOMMOP | 120 | 1.981 | 1.959 | 3.348 | 80.464 | 119.833 |
| JOMMOP01 | 250 | 2.152 | 2.176 | 2.865 | 75.562 | 114.132 |
| Fe[tren(1-nHx-2BzIm)₃](ClO4)₂ | | | | | | |
| JOMNAC | 120 | 1.963 | 1.985 | 3.441 | 80.169 | 120.092 |
| JOMNAC01 | 298 | 2.086 | 2.123 | 3.121 | 76.594 | 116.606 |
| Fe [tren(2Me-4ImH)₃]Cl(I)₃ | | | | | | |
| NAVYEP01 | 90 | 2.02 | 2.067 | 3.417 | 81.299 | 124.275 |
| NAVYEP | 180 | 2.187 | 2.202 | 3.085 | 77.041 | 116.333 |
| Fe[tren(2Thia)₃](ClO4)₂ | | | | | | |
| OSABOB01 | 100 | 1.959 | 1.951 | 3.394 | 80.937 | 119.492 |
| OSABOB | 296 | 2.142 | 2.254 | 2.837 | 74.562 | 113.107 |
| Fe[tren(5Me-4ImH)₃](ClO4)₂ | | | | | | |
| PASDOD | 100 | 1.978 | 1.972 | 3.527 | 80.997 | 119.829 |
| PASDOD01 | 293 | 2.185 | 2.206 | 3.05 | 75.665 | 115.454 |
| Fe[tren(1-nBu-2Im)₃]PF6(AsF6) | | | | | | |
| QITZIF02 | 157 | 1.969 | 1.963 | 3.311 | 80.615 | 119.496 |
| QITZIF | 230 | 2.18 | 2.222 | 2.989 | 74.996 | 113.913 |
| Fe[tren(2Me-4ImH)₃]Br(CF₃ SO₃) | | | | | | |
| ULODIJ01 | 93 | 2.037 | 2.06 | 3.228 | 79.727 | 118.606 |
| ULODIJ | 296 | 2.161 | 2.231 | 2.95 | 76.157 | 114.551 |
| Fe[tren(4Thia)₃](BF4)₂ | | | | | | |
| YAQVIY09 | 100 | 1.973 | 1.972 | 3.511 | 81.049 | 120.147 |
| YAQVIY07 | 300 | 2.155 | 2.249 | 2.809 | 74.355 | 113.524 |
| Fe[tren(2Me-4ImH)₃](PF6)Cl | | | | | | |
| YELWOC01 | 93 | 1.978 | 2.002 | 3.376 | 82.156 | 119.71 |
| YELWOC | 296 | 2.18 | 2.232 | 3.004 | 76.4 | 115.002 |
| Fe[tren(2Me-4ImH)₃](AsF6)Cl | | | | | | |
| YELWUI02 | 90 | 1.979 | 1.972 | 3.372 | 81.142 | 117.44 |
| YELWUI | 180 | 2.177 | 2.212 | 2.983 | 76.634 | 114.903 |



| Fe[tren(1Phtriazole)₃](C₂F₆ NO₄ S₂)₂ | | | | | | |
|---|---|---|---|---|---|---|
| ZOKTAW | 135 | 1.999 | 1.951 | 3.295 | 80.489 | 119.164 |
| ZOKTAW01 | 296 | 2.083 | 2.142 | 2.94 | 76.6 | 115.288 |

In general these SCO complexes are well characterized by variable temperature structures, magnetic susceptibility and possibly Mössbauer spectroscopy. The magnetic behavior for the Fe[tren(PyrzH)₃]X₂ (X⁻ = NO₃⁻, ClO₄⁻ or BF₄⁻) [34,35] salts is provided. Fe[tren(PyrzH)₃](NO₃)₂ is characterized by a fairly sharp transition at T½ = 139 K. The transition to HS is 90% complete above the SCO point. The behavior of the other two salts (ClO₄⁻ or BF₄⁻) differs in that the transitions are not sharp and the conversion to LS is incomplete even at low temperature. A few other systems that exhibit different behavior, striking counterion dependence, room temperature SCO with hysteresis, and a complex that has two HS and two LS forms, are discussed below.

An extremely interesting dependence of SCO on the counterion was observed in the [Fetren(2-Me-4ImH)₃]ClX (X= I₃⁻, PF₆⁻, SF₆⁻, SbF₆⁻, CF₃SO₃⁻) series. [Fetren(2-Me-4ImH)₃]Cl(I₃) exhibits an abrupt SCO with no hysteresis at a T½ of 110 K [36]. The structure of this complex reveals a chloride ion at the apex of a trigonal pyramid that is hydrogen bound to a N-H group of three [Fetren(2-Me-4ImH)₃]²⁺ cations resulting in a 2D layered structure. The other counterion, triiodide, exists in two independent positions, in a layer between the 2D sheets and occupying a void in the 2D layer. Replacement of the I₃⁻ (one step abrupt SCO) with PF₆⁻, SF₆⁻, SbF₆⁻, and CF₃SO₃⁻ does not result in a change to the 2D structure arrangement described above but does result in a significant change in the SCO [37]. PF₆⁻ exhibits a one step HS ⇄ (LS+HS)/2, SF₆⁻ exhibits a two step HS ⇄ (LS+HS)/2 ⇄

LS, SbF₆⁻ exhibits a gradual one step HS ⇄LS, and lastly CF₃SO₃⁻ exhibits an abrupt one step(with hysteresis) HS ⇄LS spin crossover. There are changes to the space group in these different forms but all of these mixed ion compounds are in the monoclinic P21/x family.

A SCO complex that has a spin transition near room temperature with hysteresis has potential applications. These features are found in Fetren(2Thia)₃]X₂ (X= BF₄⁻, SbF₆⁻ CF₃SO₂⁻) [38]. The complexes exhibit abrupt to gradual spin transitions near room temperature with hysteresis. The tetrafluoroborate salt has a sharp transition with a hysteresis difference between heating and cooling of 3–4 K. The above behaviors are observed even in solution. These complexes exhibit no hydrogen bonding due to the lack of an acidic N-H bond.

A SCO point of 208 K was observed for the isomeric [Fetren(4Thia)₃](BF₄)₂ [39]. In this case hysteresis was not pronounced and the spin transition was gradual with a discontinuity at 175 K. The discontinuity was investigated by determination of the structure below and above this point. It was determined that there was a crystallographic phase change from P2₁/c below 190 to P2₁ above this temperature with a corresponding trebling of Z, from 4 to 12. The surprising feature of this complex is that despite the crystallographic phase change the SCO is gradual, not abrupt. Normally a cooperative process such as a phase change is associated with an abrupt SCO. The observed discontinuity at 175 K in the SCO is attributed to an intermediate phase, which consists of mixed spin states of iron with only 30% HS.

An extremely interesting SCO complex from both a structural and magnetic standpoint is [Fetren(1-nBu-2Im)₃](PF₆)₂. Initially the complex was characterized as having two different LS forms, LS₁ and LS₂, that differed in T½, width (K) of hysteresis loop and depended on the cooling rate [40].

(T½ = 122 K, 14 K, 4.0 K/min) LS₁ ⇄ HS ⇄ LS₂ (T½ = 156 K, 41 K, 0.1 K/min)

The HS to LS₂ transition was characterized as a first order phase transition. Further study of this system using LIESST (light induced excited spin state trapping) [41] revealed the existence of two HS states [42], HS₁¹ⁱʳʳ and HS₁²ⁱʳ. HS₁¹ⁱʳʳ and HS₁²ⁱʳ were produced by irradiating LS₁ at 25 K and 90 K, respectively. The relaxation time of HS₁¹ⁱʳʳ and HS₁²ⁱʳ at 80 K is 6 min and 20 h, respectively. The fast relaxation time of HS₁¹ⁱʳʳ is attributed to the fact that the conformation of the three butyl groups in both the HS and LS forms are the same, so no conformational change is needed in relaxation. The unusually long relaxation time of HS₁²ⁱʳ is due to the reordering of the n-butyl groups of the complex. The conformations of the butyl groups in HS₁¹ⁱʳʳ and LS₁ are identical but two of the n-butyl groups

of $HS_1^{2ir}$ adopt a different conformation than that found in $LS_1$. These two butyl groups that have a different conformation in the $HS_1^{2ir}$ form and must undergo a conformational change before the complex can relax to the ground state. The long relaxation time is due to the conformational changes of two of the butyl groups, which are slow at 80 K. The metrical parameters of the iron to nitrogen parameters are consistent with the spin state assignment. The conformational issues associated with a long alkyl chain or other bulky group attached to the imidazole ring are also present in [Fetren(1-nBu-2Iim)₃](ClO₄)₂ [43], Fe[tren(1-nHx-2BzIm)₃](ClO₄)₂ [43], and Fe[tren(1Phtriazole)₃](C₂F₆ NO₄ S₂)₂ [44].

The metrical parameters of these complexes and the previously discussed [Fetren(1-nBu-Im)₃](PF₆)₂ are provided in Table 3 and were consistent with DFT predictions on the spin state. These complexes are termed metallomesogens, metal complexes "softened up" by the introduction of long alkyl substituents [43]. Introduction of alky chains to SCO complexes may allow for a melting at the crystal-liquid crystal phase transfer, which will impact the SCO point. Melting has been linked to cooperativity of the SCO [45].

<u>Iron(II) cationic complexes with a 6 membered ring, pyridine-2-carboxaldehyde</u>. There are less [Fetren(Py)₃]²⁺ complexes (49) than those of a five membered ring (83, previous section). Histogram plots of the Fe-N$_{imine}$ and Fe-N$_{pyridine}$ bond distances and N$_{imine}$ – Fe-N$_{pyridine}$ and C-N$_{ap}$- C angles were similar to those of Figures 6 and 7, respectively, but they were skewed to the LS side. Most of these compounds were of [Fetren(Py)₃]X₂ or [Fetren(6-MePy)₃]X₂ (X is a uninegative anion). The [Fetren(Py)₃]X₂ complexes were LS at all temperatures examined and the [Fetren(6-MePy)₃]X₂ complexes were SCO. The superimposed structures of HS and LS [Fetren(6-MePy)₃](PF₆)₂ are shown in Figure 9.

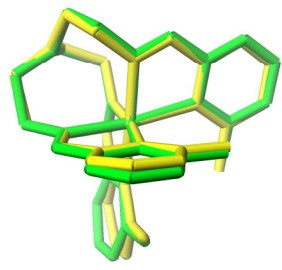

Figure 9. Superimposed structures of HS (green) and LS (yellow) [Fetren(6-MePy)₃](PF₆)₂. Note that the yellow LS form has shorter Fe-N bond distances.

The structural parameters of the LS and HS forms of [Fetren(6-MePy)₃](PF₆)₂ are as follows [46]. Fe-N$_{imine}$ (1.947 Å, 2.160 Å), Fe-N$_{pyridine}$ (2.081 Å, 2.266 Å), Fe-N$_{ap}$ (3.560 Å, 3.314 Å), N$_{imine}$–Fe-N$_{pyridine}$ (81.497°, 75.780°), and C-N$_{ap}$ –C (119.961°, 116.956°). Note that on changing from LS to HS all Fe-N bond distances increased, bond angles compressed, and the non-bonded Fe to N distance decreased. The bonded distances in this example were greater than those of Mn(III) depicted in Figure 5, as this involved a change of two electrons rather than one. The changes in the structural parameters were consistent with those from Table 2 but could not be compared directly as the calculations in Table 2 were done for a 5 membered ring aldehyde vs. the 6 membered pyridine of this example.

The placement of the methyl group into the 6 position of [Fetren(Py)₃]²⁺ caused a spin state shift from LS in [Fetren(Py)₃]²⁺ to SCO in [Fetren(6-MePy)₃]²⁺. The methyl group creates enough steric hindrance that the Fe-N$_{Pyridine}$ bond weakened and lengthened at higher temperature resulting in a cascade of other changes to the key structural parameters causing the complex to become SCO. [Fetren(6-MePy)₃](PF₆)₂ · CH₃CN · C₇H₈ [47] has the combined effects of a 6-methyl substituent, large anion and two bulky solvates. This SCO complex did not become LS until 10 K.

An interesting study of [Fetren(6–CH$_2$OH-Py)$_3$]X$_2$ (X= Br$^-$, I$^-$, BPh$_4^-$, and OTf$^-$) was done that examined the role of anion and the N$_{ap}$ atom on the SCO in the tripodal ligand [48]. It was found that as the size of the anion was increased the T$_{\frac{1}{2}}$ moved to higher temperatures. Reaction of a single arm of the above ligand (CH$_3$CH$_2$N=CH-(6-CH OHPy) =L) resulted in the HS bisterdentate complex, [FeL$_2$]X$_2$. In this case the donor set of the iron(II) is N$_4$O$_2$ rather than N$_6$. Clearly the role of the N$_{ap}$ to Fe non-bonded distance is critical in determining the spin state as was discussed in an earlier section. However, the dominant effect in this complex is the significant decrease in donor strength (N$_4$O$_2$ rather than N$_6$) that is responsible for the move to HS. Evidence of this is supported by the tris (2-pyridyl imine) N$_6$ complexes of iron(II), which are LS [49].

Placement of groups in the 5 position –CH$_2$OH [50], -C(=O)OMe [26], -OSi(CH$_3$)$_2$(t-Bu) [51], -4Py [52], -O-hexyl [53], -CF$_3$, –F, or –NO$_3$ [54] results in complexes that are LS at the temperature the structures were determined (100–170 K); but there is no structural data available at room temperature. An extremely interesting example is that of [Fetren(5-OSi(CH$_3$)$_2$(t-Bu)-Py)$_3$ ](ClO$_4$)$_2$. The silyl group is reversibly hydrolyzed by fluoride, which causes a spin state change for the iron(II) that is attached to the tren(5-OSi(CH$_3$)$_2$(t-Bu)-Py)$_3$ ligand as illustrated below.

$$Ph\text{-}O\text{-}SiR_3 + F^- \rightleftarrows Ph\text{-}O^- + SiR_3F$$

$$Fe(II) \, HS \rightleftarrows Fe(II)LS$$

While this cannot be regarded as a SCO complex, as these are two different molecules, it does allow for the use of this complex to monitor an environmental change by monitoring magnetostructural changes.

The complexes referenced in the preceding paragraph as well as [Fetren(1-nBu-2Iim)$_3$ ](ClO$_4$)$_2$, Fe[tren(1-nHx-2BzIm)$_3$](ClO$_4$)$_2$, and [Fetren(1-nBu-Im)$_3$](PF$_6$)$_2$ (mentioned in the previous section) were synthesized to prepare complexes that could be used in supramolecular systems and to examine the effect of substituents on the SCO properties of metallomesogens [48].

Iron(III) cationic complexes with a 5 membered ring carboxaldehyde. There are only four reported compounds of a tripodal tren Schiff base iron(III) cationic complex compared to 132 for the analogous iron(II) cationic complexes. This points out the apparent redox instability of the iron(III) cationic species as was mentioned earlier. Studies on these tripodal ligands have shown that there is a correlation of protonation state with oxidation state [55]. The complexes are [Fetren(4ImH)$_3$](BF$_4$)$_3$·H$_2$O· C$_4$H$_4$N$_2$O [56] (C$_4$H$_4$N$_2$O is imidazole-4-carboxaldehyde, a component of the reaction mixture) [Fetren(4ImH)$_3$](ClO$_4$)$_3$·H$_2$O· C$_4$H$_4$N$_2$O [57], [Fetren(4ImH)$_3$]Cl(PF$_6$)$_2$·H$_2$O [55], and [Fetren(5Me-4ImH)$_3$ ](ClO$_4$)$_3$·1.5 C$_2$H$_6$O(ethanol) [58]. The latter three are HS as supported by the average structural parameters: Fe-N$_{imine}$ (2.087 Å) Fe-N$_{imidazole}$ (2.076 Å), Fe-N$_{ap}$ (2.725 Å), N$_{imine}$ –Fe-N$_{imidazole}$ (76.812°), and C-N$_{ap}$ –C (113.798°). [Fetren(4-ImH)$_3$ ](BF$_4$)$_2$·H$_2$O· C$_4$H$_4$N$_2$O is SCO as supported by the following structural data. Values are given for LS (93 K) followed by HS (296 K). Fe-N$_{imine}$ (1.974, 2.060 Å) Fe-N$_{imidazole}$ (1.941, 2.090 Å), Fe-N$_{ap}$ (3.170, 2.763 Å), N$_{imine}$–Fe-N$_{imidazole}$ (80.992, 76.865°), and C-N$_{ap}$–C (118.838, 114.830°). The crystallization of [Fetren(4ImH)$_3$ ](BF$_4$)$_3$·H$_2$O· C$_4$H$_4$N$_2$O with a molecule of imidazole-4-carboxaldehyde may not be entirely happenstance. The hydrogen bonding network between the [Fetren(4ImH)$_3$]$^{3+}$ cation and the C$_4$H$_4$N$_2$O and water and anion may help stabilize the Fe(III) N$_6$ cation, which is prone to reduction. In all of these complexes the electrostatic attraction of the iron(III) cation exerts a greater pull on N$_{ap}$ resulting in a shorter Fe-N$_{ap}$ distance than observed in the iron(II) cationic species.

Iron(III) neutral complexes. There are only eleven iron(III) structures of neutral tripodal complexes of tren, eight with a N$_6$ donor set (one structure was reported twice in database and omitted from Table 4) and three with a N$_3$O$_3$ donor set. The former all contain three deprotonated imidazole carboxaldehydes condensed to tren and the latter contain three salicylaldehydes condensed to tren. The structural parameters from Table 2 for these systems are given in Table 4. The iron(III) N$_6$ complexes are LS while the iron(III) N$_3$O$_3$ complexes are HS. The change from HS iron(III) N$_6$ cationic complexes (discussed above) of a H$_3$L ligand to LS iron(III) complexes of L$^{3-}$ is explained by the field strength of the ligand. A change in spin state is anticipated since of Δ (L$^{3-}$) > Δ (H$_3$L). The HS nature of the iron(III) N$_3$O$_3$ complexes is understood in the sense that a N$_4$O$_2$ donor set is usually required

to promote LS or SCO iron(III) [59]. Since $\Delta$ (N$_4$O$_2$) > $\Delta$ (N$_3$O$_3$) the latter are anticipated to be HS if the former are LS or SCO. The experimental parameters from Table 4 for the iron(III) N$_6$ neutral complexes are consistent with the theoretical values from Table 2 and an LS assignment.

**Table 4.** Structural data for iron(III) neutral complexes. Distances are in Å and angles in (°).

| CCDC Code | Complex | Fe-N$_{imine}$ | Fe-N | Fe-N$_{ap}$ | N$_{imine}$-Fe-N (O) | C-N-C |
|---|---|---|---|---|---|---|
| BAKSOV | Fetren(5Me-4Im)$_3$ | 1.987 | 1.941 | 3.346 | 81.108 | 119.016 |
| DEYNEB | Fetren(2Im)$_3$ [a] | 1.988 | 1.933 | 3.194 | 81.059 | 118.559 |
| EYIHUP | Fetren(4Im)$_3$ | 1.988 | 1.938 | 3.257 | 80.959 | 119.004 |
| EYIJAX | Fetren(2Im)$_3$ [b] | 1.99 | 1.933 | 3.186 | 80.966 | 119.269 |
| FEJBIG | Fetren(4Im)$_3$·HPF$_6$ | 1.987 | 1.953 | 3.165 | 81.116 | 117.93 |
| IMANAL | Fetren(2Me-2Im)$_3$ | 2.03 | 2.001 | 3.346 | 80.063 | 119.595 |
| TAZBFE | Fetren(Pyr)$_3$ | 1.99 | 1.936 | 3.304 | 81.041 | 119.467 |
| average | | 1.992 | 1.944 | 3.254 | 80.875 | 118.980 |
| CESALF10 | Fetren(5ClSal) | 2.185 | 1.954 | 3.261 | 85.6 | 117.837 |
| KENXOS | Fetren((4-5-MeS)PhSal)$_3$ | 2.174 | 1.94 | 3.353 | 87.493 | 119.271 |
| ZOHMOY | Fetren(Sal)$_3$ | 2.174 | 1.976 | 3.127 | 87.166 | 113.839 |
| average | | 2.177 | 1.956 | 3.247 | 86.753 | 116.982 |

[a] SG R-3c; [b] SG P21/c.

Effect of protonation of the imidazole ring with iron(III) N$_6$ complexes. As was discussed in the previous sections the tripodal [Fetren(ImH)$_3$]$^{3+}$ cations are HS but the deprotonated Fetren(Im)$_3$ complexes are LS. This cannot be regarded as SCO as these are not the same molecules. However, it is a change in the spin state that is due to a rapid reversible chemical reaction, protonation/deprotonation. The data support a spin state change that is affected, not by a change in temperature or pressure, but an environmental condition, pH. The structural effects of protonation/deprotonation are illustrated in Figure 10 for [Fetren(5Me-4ImH)$_3$]$^{3+}$ [58] and Fetren(5Me-4Im)$_3$ [60]. There are potential applications of this spin state change based on a simple rapid and reversible reaction. There are other simple reactions of this type (exposure to dioxygen or CO) that could be used with other complexes to affect a spin state change [12, 13].

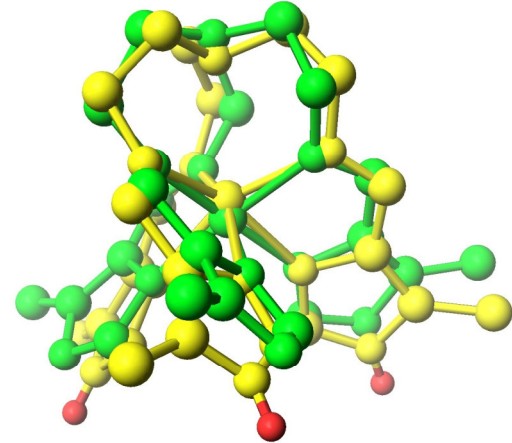

**Figure 10.** Superimposed structures of HS (yellow) [Fetren(5-Me-4-ImH)$_3$]$^{3+}$ and LS (green) [Fetren(5-Me-4-ImH)$_3$]. All hydrogen atoms have been removed for clarity except the three attached in the protonated form (shown in red). The structural parameters of these complexes, Fe-N$_{imine}$, Fe-N$_{imidazole}$,

Fe-N$_{ap}$, N$_{imine}$ –Fe-N$_{imidazole}$, and C-N$_{ap}$–C are as follows: HS(yellow) [Fetren(5Me-4ImH)$_3$]$^{3+}$ 2.106 Å, 2.163 Å, 2.589 Å, 75.915°, and 111.506° and LS(green) [Fetren(5Me-4Im)$_3$] 1.988 Å, 1.938 Å, 3.257 Å, 80.959°, and 119.004°.

Cobalt complexes. This discussion is limited to cobalt(II) (d$^7$) systems as all octahedral (O$_h$) cobalt(III) complexes, other than those of fluoride, are LS. There are several reviews [61–63] of SCO cobalt(II) complexes as well as a report of a reverse spin transition [64]. SCO for Oh cobalt(II) requires a N$_6$ donor set and the majority of systems studied are of phenathroline, terpyridine, and closely related ligands. The average of the reported Co-N bond length changes from LS to HS is 0.10Å. A SCO has also been observed in the HS Co(bipy)$_3$ $^{2+}$ cation when crystallized in a 3D oxalate network [65].

There are a relatively small number of mononuclear tripodal cationic cobalt(II) complexes of tren and these are given in Table 5 with their structural parameters. The structural data support a HS assignment in all cases but [Cotren(2ImH)$_3$](ClO$_4$)$_2$ [66], which has the structural signature of LS, which suggest it may undergo a SCO above room temperature. Only structural data has been reported on this system and therefore it is quite possible that this is not a cobalt(II) complex at all. Reaction of tren(2ImH)$_3$ with cobalt(II) perchlorate in air produces [Cotren(2ImH)$_2$(2Im)](ClO$_4$)$_2$ [67], which has structural parameters nearly identical to the proposed [Cotren(2ImH)$_3$](ClO$_4$)$_2$. The difference in formula between these two systems is a single hydrogen atom but the implications are significant, cobalt(II) vs. cobalt(III). In addition the space groups of [Cotren(2ImH)$_3$](ClO$_4$)$_2$ (P21/n) and [Cotren(2ImH)$_2$(2Im)](ClO$_4$)$_2$ (P21/c) are similar and the cell parameters of the two salts are about the same except that the c value of [Cotren(2ImH)$_3$](ClO$_4$)$_2$ is double the size of [Cotren(2ImH)$_2$(2Im)](ClO$_4$)$_2$. Below 77 K structure determinations were attempted on [Cotren(1Me-2Im)$_3$](ClO$_4$)$_2$ to see if there was evidence of a LS state but the complex underwent a phase change and it was not possible to solve for a structure [68].

**Table 5.** Structural parameters of mononuclear cationic cobalt(II) complexes. Distances are in Å and angles in °.

| Complex | Co-N i$_{mine}$ | Co-N(O) | Co-N$_{ap}$ | N $_{imine}$-Co-N(O) | C-N$_{ap}$-C |
|---|---|---|---|---|---|
| [Cotren(5MeCO2Py)$_3$] (CoCl$_4$)$_2$ | 2.079 | 2.228 | 2.626 | 74.495 | 113.104 |
| [Cotren(py)$_3$] (ClO$_4$)$_2$ | 2.119 | 2.17 | 2.884 | 76.121 | 114.574 |
| [Cotren(5-tBuNHCO-2Py)$_3$]Cl$_2$ | 2.121 | 2.262 | 2.574 | 74.083 | 111.623 |
| [Cotren(5tBuNHCO-2Py)$_3$](ClO$_4$)$_2$ | 2.11 | 2.264 | 2.706 | 74.333 | 113.752 |
| [Cotren(5tBuNHCO-2Py)$_3$]I | 2.105 | 2.258 | 2.633 | 74.578 | 112.356 |
| [Cotren(5-tBuNHCO-2Py)$_3$]Br$_2$ | 2.107 | 2.253 | 2.592 | 74.596 | 111.809 |
| [Cotren(py)$_3$](BF$_4$)$_2$ | 2.091 | 2.227 | 2.87 | 75.422 | 113.719 |
| [Cotren(pyrz)$_3$](BF$_4$)$_2$ | 2.08 | 2.346 | 2.503 | 72.703 | 110.887 |
| [Cotren(N-oxyPy)$_3$](PF6)2 | 2.149 | 2.162 | 2.443 | 78.768 | 111.715 |
| [Cotren(2Ph-2ImH)$_3$](NO$_3$)$_2$ | 2.133 | 2.237 | 3.267 | 78.068 | 116.627 |
| [Cotren(1Me-2ImH)$_3$](ClO$_4$)$_2$ | 2.131 | 2.164 | 2.946 | 76.18 | 114.819 |
| [Cotren(5Me-4ImH)$_3$](ClO$_4$)$_2$ | 2.124 | 2.205 | 2.72 | 75.627 | 113.212 |
| [Cotren(2ImH)$_3$](ClO$_4$)$_2$ | 1.914 | 1.862 | 3.331 | 82.188 | 119.51 |

## 4. Supramolecular Systems

Supramolecular chemistry [69–71] involves the self-assembly of small building blocks to form larger molecular arrays. It is often dependent on the shapes of the smaller molecules or hydrogen bonding and can involve elements of chiral recognition, size of a component, or a host–guest relationship. The ability of the tripodal SB tren(azoleH)$_3$ complexes to form supramolecular assemblies is something that distinguishes this class of SCO complexes. In the case of the tren(azoleH)$_3$ complexes it is predominately hydrogen bonding that is responsible for the formation of supramolecular assemblies. An Mtren(azoleH)$_3$ $^{2+\ or\ 3+}$ [the two possible charges in this and the following examples are due to the fact that the M can be either a M(II) or a M(III)] has three hydrogen

bonding donor sites and an Mtren(azole)$_3^{1-\text{ or }0}$ complex has three hydrogen bonding receptors. This results in a natural driving force for an interaction to form a larger assembly. The donor/acceptor interaction could be as given above or could be from identical hemideprotonated species, [Mtren(azoleH$_{0.5}$)$_3$]$^{0.5+\text{ or }1.5+}$. The hemideprotonated species is possible if the three arms of the tripodal ligand are crystallographically identical and the occupancy of the azole hydrogen atom is 0.5. Regardless of the precise chemical identity of the interacting species, Mtren(azoleH)$_3$ $^{2+\text{ or }3+}$ with Mtren(azole)$_3^{1-\text{ or }0}$ or two identical Mtren(azoleH$_{0.5}$)$_3$ $^{0.5+\text{ or }1.5+}$, the overall result is the same, two complexes with three hydrogen atoms and three donor/acceptor sites. The only distinction depends on whether or not the metal complexes are crystallographically distinguishable. In a supermolecular system both metals can be the same (homonuclear) or different (heteronuclear). They can have the same or different oxidation states or an average oxidation state (M$^{2.5+}$) as found in intervalent [72] compounds such as Fe$_3$O$_4$ or basic iron acetate, Fe$_3$O(OAc)$_6$. In addition, both or one of the metals may be a spin crossover. The level of aggregation of these molecular building blocks, Mtren(azoleH$_{0.5}$)$_3$ $^{0.5+\text{ or }1.5+}$, depends upon the spatial relationship of the nitrogen atoms in the five membered ring. If they have a 1,2 relationship, as in pyrazole, discrete dinuclear complexes result; and if the relationship is 1,3, as in imidazole, then a 2D sheet of hexagons or a tetrahedral tetranuclear complex results.

<u>Dinuclear complexes</u>. These are prepared by an aerial oxidation of Fetren(5-MePyrzH)$_3^{2+}$ to give {[Fetren(5-MePyrzH)$_2$(5-MePyrz)][Fetren(5-MePyrz)$_2$(5-MePyrzH)]}X$_2$ [73], reaction of Mtren(5-MePyrzH)$_3^{2+}$ (M = Fe(II) or Mn(II)) with Cotren(5Me-Pyrz)$_3$ to give {[Mtren(5Me-PyrzH)$_3$][ Cotren(5-MePyrz)$_3$ ]}X$_2$ [74], or protonation of a Cotren(5MePyrz)$_3$ to give {[Cotren(5-MePyrzH$_{0.5}$)$_3$]$_2$ X$_3$ (X$^-$=ClO$_4^-$ or BF$_4^-$) or {[Cotren(5-MePyrzH)$_3$][Cotren(5-MePyrz)$_3$]}I$_2$ [75]. Regardless of the metal or oxidation state the complexes exhibit three structural features in common. (1) The dimers have three N$_{\text{pyrazole}}$$\cdots$H$\cdots$ N$_{\text{pyrazolate}}$ hydrogen bonds that link the two halves together. (2) The dimers exhibit $\pi$–$\pi$ stacking of the three pairs of pyrazole rings. (3) Homochirality of both metal complexes of the dimer/pseudodimer, either both $\Delta$ or both $\Lambda$ is essential (chiral recognition) for the two halves to interlock. These features are illustrated in Figure 11 for {[Fetren(5-MePyrzH)$_3$][Fetren(5-MePyrz)$_3$]}$^{2+}$. The magnetostructural data support that the iron(II) ion is SCO and weekly coupled to the iron(III) component.

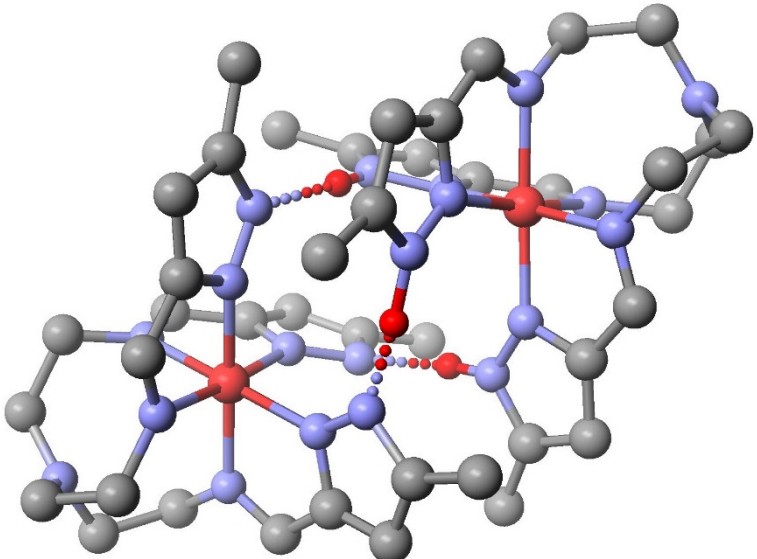

**Figure 11.** Structure of the {[Fetren(5-MePyrzH)$_3$][Fetren(5-MePyrz)$_2$]}$^{2+}$ cation. Hydrogen atoms have been omitted for clarity except for those involved in N$_{\text{pyrazole}}$–H$\cdots$ N$_{\text{pyrazolate}}$ bonding, shown in red. The iron(II) complex (upper right) is SCO. Fe-N$_{\text{imine}}$ Fe-N$_{\text{pyrazole}}$, Fe-N$_{\text{ap}}$, N$_{\text{imine}}$–Fe-N$_{\text{pyrazole}}$, and C-N$_{\text{ap}}$–C for iron(II) at 123 K are as follows: 1.9741 Å, 1.9552 Å, 3.446 Å, 79.77°, and 120.0°.

2D sheets. The above complex was of a mixed valence iron(II)–iron(III) spin crossover molecule that was a discrete dimer due to the positional location of the nitrogen atoms (positions 1 and 2) in the pyrazole ring. Changing the positions of the nitrogen atoms to 1 and 3 yields complexes exhibiting the same 1:1 stoichiometry, but the overall topology is radically different. In this case, the supramolecular feature is a 2D sheet. Reaction of Fetren(4ImH)$_3^{2+}$ with Fetren(4Im)$_3$ gives a polymer of {[Fetren)(4-ImH)$_3$][Fetren(4-Im)$_3$]}X$_2$ [76,77] (X$^-$ = NO$_3^-$ or BF$_4^-$). The same product is produced by te reaction of 1.5 equivalents of base with Fetren(4ImH)$_3$ $^{2+}$. The base forms the hemideprotonated species and half of the iron(II) is oxidized to iron(III) in air. It should be pointed out that the synthesis of Fetren(4ImH)$_3^{2+}$ was from the reaction of an iron(III) salt with the ligand. This unusual aerial reduction of the hypothetical Fetren(4ImH)$_3^{3+}$ to iron(II) has been pointed out earlier in this review and points to the general instability of the iron(III) N$_6$ cationic species.

There are three similarities between the dinuclear complexes discussed above with the present series. Both are a 1:1 iron(II) cationic complex with an iron(III) neutral of the same ligand, both metals are of the same chirality, Δ or Λ, and both can form three hydrogen bonds as on average they are hemideprotonated. The topologies differ drastically as the tren(5-MepyrzH$_{0.5}$)$_3$ complex can form three hydrogen bonds to another molecule of itself but the tren 4-imidazole complex forms three hydrogen bonds to three different complexes resulting in a 2D sheet structure shown in Figure 12. Each hexagon in Figure 12 contains three iron(II) cations, [Fetren)(4-ImH)$_3$]$^{2+}$ (H donor), and three iron(III) neutral [Fetren(4-Im)$_3$] complexes (H acceptor). Each complex forms two hydrogen bonds to an adjacent iron complex in the same ring and one to a complex in an adjoining ring.

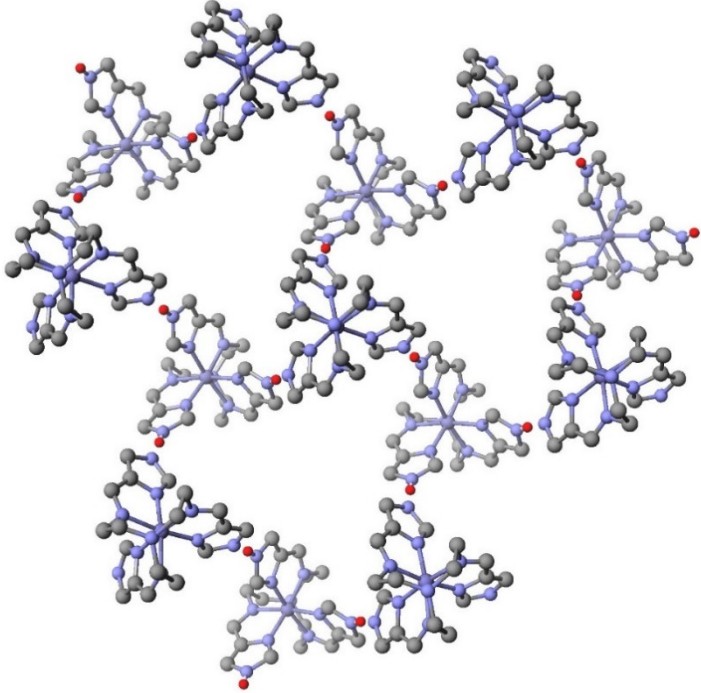

**Figure 12.** Structure of a portion of the polymeric 2D sheet of {[Fetren)(4-ImH)$_3$][Fetren(4-Im)$_3$]}(BF$_4$)$_2$ at 100 K. The counteranions and all hydrogen atoms except those participating in N$_{imidazole}$ − H$^{\cdots}$N$_{imidazolate}$ bonding (shown in red) have been omitted for clarity. Note the repeating hexagonal groupings of iron complexes.

This complex undergoes the following spin crossovers LS Fe(II) LS Fe(III) to HS Fe(II) LS Fe(III) to HS Fe(II) HS Fe (III). These conclusions were based on a combination of variable temperature magnetic susceptibility measurements and Mössbauer spectroscopy. Mössbauer spectroscopy showed that the iron(II) underwent its SCO at lower temperature than the iron(III). At room temperature only a fraction of the iron(III) had undergone the SCO. The various percentages of HS and LS iron(II) and iron(III) over the temperature range examined were consistent with the magnetic susceptibility data.

Structures at different temperatures were determined as well, but even at 300 K the iron(III) is only partially converted to HS, which is supported by the magnetic susceptibility measurements and Mössbauer studies. The structural data for this system is more complex than many SCO complexes as on lowering the temperature one of the unit cell parameters is halved. This is a truly remarkable system that has undergone further study [78]. Structural parameters for the iron(II)-iron(III) 2D polymer, {[Fetren)(4-ImH)$_3$][Fetren(4-Im)$_3$]}(BF$_4$)$_2$, and the iron(II)-iron(II) polymer (discussed next) at several temperatures are given in Table 6. The agreement with values from Table 2 is very good.

**Table 6.** Structural data for {[Fetren(2Me-4ImH)$_3$][Fetren(2Me-4Im)$_3$]}NO$_3$ and {[Fetren)(4-ImH)$_3$][Fetren(4-Im)$_3$]}(BF$_4$)$_2$ 2D sheet polymers.

| Component | T | Fe-N$_{imine}$ | Fe-N$_{imid}$ | Fe-N$_{ap}$ | N-$_{imine}$-Fe-N-imid. | C-N-$_{ap}$-C |
|---|---|---|---|---|---|---|
| Fetren(2Me-4Im)$_3$$^-$ | 90 K | 2.03 | 2.059 | 3.366 | 79.998 | 119.316 |
| | 130 K | 2.066 | 2.095 | 3.28 | 79.43 | 118.248 |
| | 293 K | 2.171 | 2.184 | 3.122 | 77.004 | 115.393 |
| Fetren(2Me-4ImH)$_3$$^{2+}$ | 90 K | 2.066 | 2.082 | 3.371 | 79.575 | 119.424 |
| | 130K | 2.077 | 2.107 | 3.314 | 78.732 | 119.067 |
| | 293 K | 2.173 | 2.199 | 3.169 | 76.932 | 116.777 |
| Fetren(4Im)$_3$ | 100 K | 1.988 | 1.947 | 3.302 | 80.908 | 119.43 |
| | 220 K | 2.021 | 2.005 | 3.143 | 79.667 | 117.836 |
| | 300 K | 2.013 | 1.952 | 3.202 | 80.82 | 118.388 |
| | | | | | | |
| Fetren)(4ImH)$_3$$^{2+}$ | 100 K | 1.986 | 1.951 | 3.314 | 81.034 | 119.56 |
| | 220 K | 2.023 | 2.008 | 3.157 | 79.755 | 118.277 |
| | 300 K | 2.026 | 1.979 | 3.16 | 79.642 | 118.086 |

Following on the remarkable success of the previous system, this methodology using the tripodal tren(2Me-4ImH)$_3$ ligand [79,80] was extended to iron(II)-iron(II) 2D sheets. The repeating species is {[Fetren(2Me-4ImH)$_3$][Fetren(2Me-4Im)$_3$]}$^+$. This cation was isolated and characterized with six different anions. While the [Fetren(2Me-4Im)$_3$]$^-$ is clearly a component of the above polymer there does not appear to be a simple M$^+$ salt of any [Fetren(4Im)$_3$]$^-$ or [Fetren(2Im)$_3$]$^-$ for structural or magnetic characterization. However, it is well characterized in these 2D sheets. The compound undergoes a two step SCO, LS Fe(II) cation LS Fe(II) anion to HS Fe(II) cation LS Fe(II) anion and lastly to HS Fe(II) cation HS Fe(II) anion. The iron(II) anion undergoes its SCO at higher temperatures as the L$^{3-}$ ligand is a stronger donor than the neutral H$_3$L. Subsequent work on the parent complex, [Fetren(2-Me-4-ImH)$_3$]X$_2$ revealed dependence of the spin state and SCO temperature on the identity of the anion [81].

The previous two SCO systems were 2D sheets of Fe(II)-Fe(III) and Fe(II)-Fe(II). Fe(III)-Fe(III) systems are produced from the hemideprotonated [Fetren(5Me-4ImH)$_3$]$^{1.5+}$ cation [82]. In this case four complexes are produced, three having the formula, [Fetren(5Me-4ImH)$_3$](ClO$_4$)$_{1.5}$·(H$_2$O)$_x$, with three different values of x. All are of the 2D sheet structure. The fourth complex is the anhydrous [Fetren(5Me-4ImH)$_3$](ClO$_4$)$_{1.5}$, which is depicted in Figure 13. These examples illustrate the remarkable structures that are possible with the [Mtren(azoleH$_{0.5}$)$_3$]$^{0.5, 1.0 \text{ or } 1.5+}$ cation as a supramolecular building block. If pyrazole is used dinuclear complexes result and if imidazole is used two results are possible, 2D sheets and a tetranuclear cluster. In all cases these species are homochiral at the level of the molecule or perhaps the entire crystal if they crystallize in a Sohncke space group [83]. In addition this allows for the inclusion of SCO, intervalent, and spin exchange behavior all in the same easy to prepare molecular system.

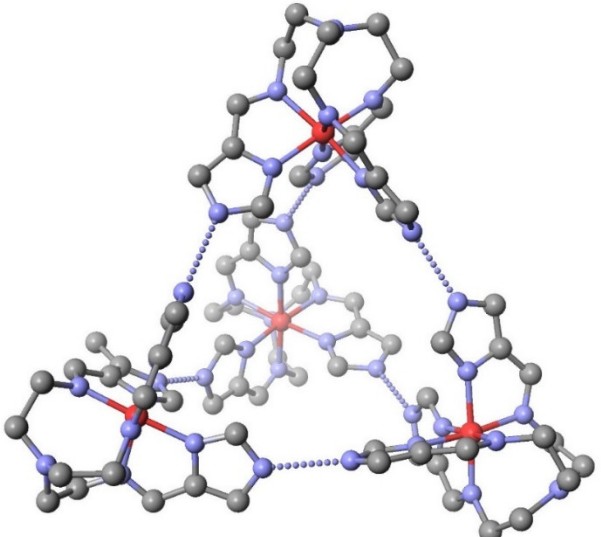

**Figure 13.** The supramolecular structure of [Fetren(5Me-4ImH$_{0.5}$)$_3$](ClO$_4$)$_{1.5}$. All hydrogen atoms and methyl groups have been removed for clarity. There are six N$_{imidazole}$–H$\cdots$N$_{imidaxolate}$ hydrogen bonds shown as a dashed line. Fach complex forms three hydrogen bonds to each of the other three complexes. The iron atoms are on the vertices of a perfect tetrahedron.

There are also structures of heteronuclear 2D sheets using the tripodal imidazole ligands. These include [Fe(III) H$_3$ L][CoL](ClO$_4$)$_3$ [82], a Cu(II)-Fe(III) [84], and an Mn(II)-Fe(III) [85] system but the limited magnetic characterization of these complexes, unlike the earlier 2D sheet structures does not allow for any SCO conclusion.

Double salts. Reaction of [Fetren(2ImH)$_3$]X$_2$ (X= ClO$_4^-$ or BF$_4^-$) with MX (M = NH$_4^+$, Na$^+$, K$^+$, Rb$^+$ or Cs$^+$) in methanol gives nine isomorphous double salts, [Fetren(2-ImH)$_3$]MX$_3$ [86,87]. These are supramolecular systems held together by hydrogen bonding and electrostatic attraction. A single M$^+$ is surrounded by six X$^-$ anions which in turn hold six [Fetren(2-ImH)$_3$]$^{2+}$ cations in place through extensive hydrogen bonding involving both N$_{imidazole}$-H$\cdots$X and N=C-H$\cdots$X interactions. The parent [Fetren(2-ImH)$_3$]$^{2+}$ is HS at room temperature but becomes LS in the double salts. The change in spin state was followed by Mössbauer spectroscopy. This change in ground state is due to the fact that the three N$_{imidazole}$-H bonds in the double salts act as donors to the oxygen and fluorides of the perchlorate and tetrafluoroborate anions. This shifts the ImH group closer to an imidazolate, which increases the strength of the ligand and results in the change of spin state. While not a true SCO system it is much like the HS [Fetren(4-ImH)$_3$]$^{3+}$ and LS [Fetren(4-ImH)$_3$] system discussed earlier in the protonation section. In the iron(II) double salt series you have weakening of a N$_{imidazole}$–H bond and with the iron(III) in [Fetren(4-ImH)$_3$]$^{3+}$ you have breakage of the same bond. In both cases weakening of the N$_{imidazole}$–H bond results in a spin state change from HS to LS. Both of these systems could be used in a materials application as the spin state is reversibly changing due to environmental considerations, which alters both the magnetic and optical properties of the complex.

Supramolecular system of a substituted iron(II) tren(py)$_3$ complex. The above examples all employ tren(Im)$_3$ or tren(ImH)$_3$ complexes but there are examples with pyridine as well. Reaction of two equivalents of [Fetren(5(-4pyridyl)Py)$_3$]$^{2+}$ with three equivalents of [Pd(dppe)]$^{2+}$ (dppe = (1,3 bisdiphenylphosphino)propanoyl) resulted in the eloquent and rationally assembled supramolecular complex [**Error! Bookmark not defined.**] pictured in Figure 14. The reaction is based on the self assembly of building blocks based entirely on the shape of the blocks. Each of the two tripodal (C$_3$ symmetry) [Fetren(5(-4-pyridyl)Py)$_3$]$^{2+}$ complexes has three long tails that terminate with a pyridine nitrogen atom (Lewis base). The [Pd(dppe)]$^{2+}$ ion (C$_2$ symmetry) has two vacant coordination positions (Lewis acids). The Lewis acid base reaction between [Pd(dppe)]$^{2+}$ and [Fetren(5(-4-pyridyl)Py)$_3$]$^{2+}$ gives the eloquently formed trigonal bipyramidal complex.

{[Fetren(5(-4-pyridyl)Py)$_3$]$_2$ [Pd(dppe)]$_3$}$^{10+}$ is depicted in Figure 14. This supramolecular complex is held together by coordinate covalent bonds unlike all the other systems, which utilize hydrogen bonding.

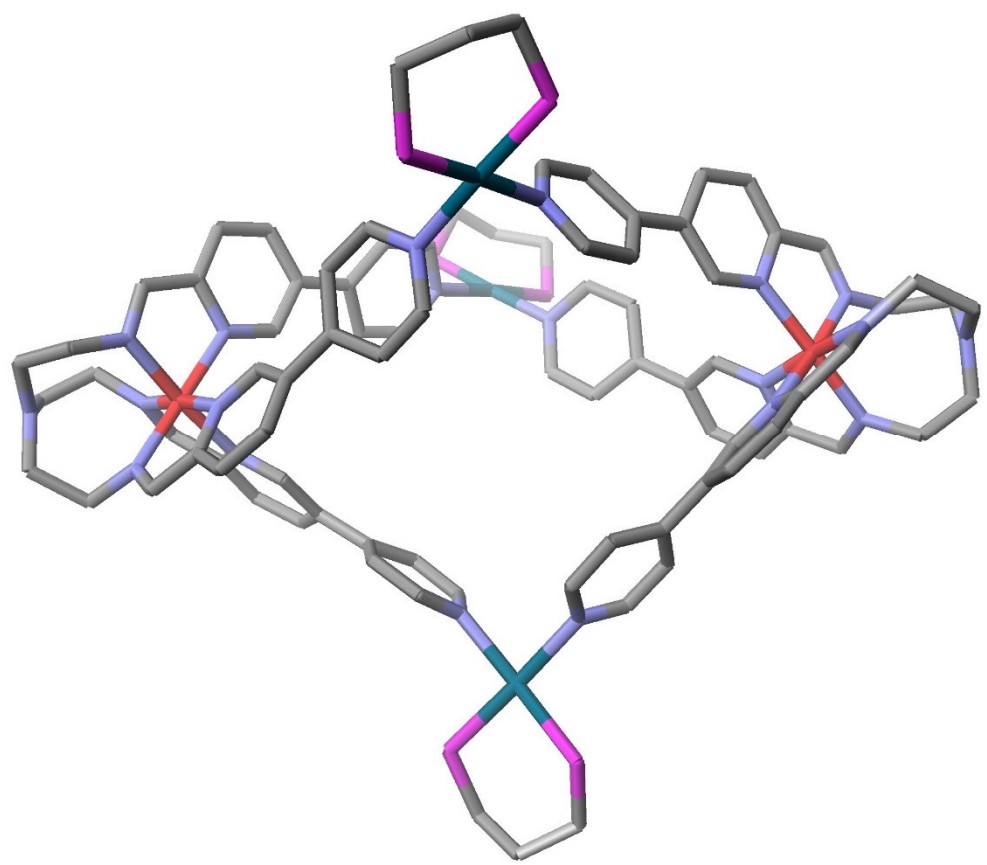

**Figure 14.** A Fe$_2$Pd$_3$ supramolecular complex, {[Fetren(5(-4-pyridyl)Py)$_3$]$_2$ [Pd(dppe)]$_3$}$^{10+}$, where the iron atoms occupy the axial sites and the palladium atoms the equatorial sites. All hydrogen atoms, phenyl groups of the dppp ligand, anions, and solvent have been omitted for clarity. The structural parameters of the iron(II) are consistent with a LS assignment.

## 5. Summary and Perspective

A review of the Cr, Mn, Fe, and Co complexes of tripodal Schiff bases prepared by the condensation of tren (N$_{ap}$(CH$_2$CH$_2$NH$_2$)$_3$) or related backbones with various aldehydes that contain a nitrogen or oxygen donor has been presented. The key focus was on metrical parameters that correlate with spin state selection and/or a SCO. Four conformations of the backbone, "N$_{ap}$ out", "Planar", "N$_{ap}$ in", and "extreme N$_{ap}$ in" are observed in these complexes. The distance between the metal and N$_{ap}$ decreases in the order the terms are given. "N$_{ap}$ out" and "extreme N$_{ap}$ in" are locked into LS and HS states respectively and are not involved in SCO with any known examples. SCO involves equilibrium between "Planar" and "N$_{ap}$ in", "Planar" ⇄ "N$_{ap}$ in". As the temperature increases a LS complex shifts to HS. The motion of the ligand responsible for this is like turning a screw (N$_{ap}$) in towards the metal. This causes the metal to N$_{ap}$ distance to decrease, the C-N$_{ap}$–C angles to decrease, and three N$_{imine}$ atoms to bend back lengthening their bonds. As the three N$_{imine}$ atoms move away from the metal they are pushed closer to the other three donor atoms. This results in a narrowing of the bite angle and a lengthening of the other three metal donor distances. All five of these structural parameters move in unison during the LS to HS conversion. Theoretical predictions for the values of these five parameters are provided in Table 2.

An examination of the literature for mononuclear SCO complexes of Cr and Mn reveals only one clear example, Mntren(pyr)$_3$. A structural search of the cationic mononuclear Fe(II) N$_6$ complexes containing a five membered ring or six membered ring resulted in 83 and 49 hits, respectively.

Histogram plots of the key structural parameter displayed bimodal behavior indicating two different electronic ground states, HS and LS. The values for the two maxima in the histogram plots (bond distances and bond angles) agreed well with earlier theoretical predictions. Specific compounds that exhibited unusual aspects of their SCO were discussed. Far less structural data was available on the iron(III) cationic complexes. This correlates with synthetic observations that many iron(III) cationic complexes of this class reduce to iron(II) in air. A clear example of one SCO cationic iron(III) was discussed. The $N_6$ and $N_3O_3$ neutral iron(III) complexes were all LS and HS respectively. All cobalt(II) complexes were HS.

There were numerous examples of supramolecular complexes of iron that had the features of chiral recognition, intervalence, and SCO all in a single system. These systems self assemble largely through hydrogen bonding to give dimers, tetrahedral clusters, and 2D sheets. The shape of the system was determined in large part on the aldehyde component, imidazole, or pyrazole.

This is clearly an interesting and important class of SCO complexes with two dozen SCO mononuclear complexes characterized and a nearly equal number of supramolecular systems that are extremely unusual. Despite this there are missing pieces. On the ligand side there are no structurally characterized $N_5O$, $N_4O_2$ complexes (they have been synthesized), and there are no structurally characterized $N_3S_3$ complexes but a tren $N_3S_3$ free ligand has been structurally characterized. Altering the donor strength gives the chemist the ability to fine tune the ligand field strength. To date there are no mononuclear Cr(II), Mn(II), or Co(II) SCO complexes known despite the fact that ligands of this class form stable complexes with these metals. Perhaps the most glaring omission is the lack of other Mn(III) SCO complexes. Since the Mn(III) of [Mntren(pyr)$_3$] is SCO in a $N_6$ donor set there is reason to believe that [Mntren(4Im)$_3$], [Mntren(2Im)$_3$] or their simple alkyl substituted versions should be SCO as well. To date these complexes have not been reported. There has also been only a small amount of supramolecular work done with these metals. Given the success with Fe(II), Fe(III), and Co(III), it is reasonable that these other metals may form the same type of systems.

**Supplementary Materials:** The following are available online at www.mdpi.com/2312-7481/6/2/28/s1.

**Funding:** This work was partially supported by NASA, Goddard Space Flight Center, NASA NNX15AM13A.

**Conflicts of Interest**: The authors declare no conflict of interest.

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
