# Peer review of "Structural Evidence of Spin State Selection and Spin Crossover Behavior of Tripodal Schiff Base Complexes of tris(2-aminoethyl)amine and Related Tripodal Amines"

_magnetochemistry, doi:10.3390/magnetochemistry6020028_

Round 1

Reviewer 1 Report

G. Brewer gives a review on structural aspects of tripodal Schiff base complexes featuring spin crossover.
The manuscript is in the present form not suitable for publication. I have several general issues I request to consider, as well as specific ones throughout the manuscript.

General
- The manuscript is missing a clear structure, starting from the abstract with giving the reader a clear picture what he will learn by reading the review.
- I miss the emphasis on the magnetic aspect of the spin crossover review, as e.g. not a single magnetization curve was represented. As the article was submitted to magnetochemistry, this should be a central aspect. If only structural considerations are the subject of the article, probably a journal as Crystals would be better suited.
- Please explain all abbreviations throughout the article the first time they are used
- The complete article demands for a careful proof-reading. I assume, the author is a native speaker. Nevertheless, there are lots of typos needing corrections, several issues with punctuation (often missing, some superfluous) and phrasing. Please use short sentences and avoid nested sentences, especially when commas are missing. I suggest also using past tense and avoiding the repetitive use of future (e.g. “there will be…”, etc.)
- Of course, it is impossible to mention and describe all examples covered by the topic of a review article. Nevertheless, it is of utmost importance to explain the reader carefully why the selected examples were selected, or why they are considered as important. This step is missing here
- Between all figures and unit symbols (even %) a blank is required
- A proof-reading by a native speaker to improve the language-mistakes (tense, plural/singular, phrasing, terminology, …) as this would enhance the readability of the manuscript
- For all single items / compounds in the tables the references are needed in the tables
- All statements (and there are a few in the manuscript) that there would be more work necessary should be removed. A review article is a representation of a status quo, not a personal interpretation of what should or could be done. If there is the urgent need to point out insights from this review, which could be suitable to advance the field, than those should be grouped in a single section at the end with a proper justification.
- All structures should be given as ORTEP-plots (or have a sound explanation why a different representation was chosen), have a short description what exactly is shown and the reader should see, as well have the proper reference in the figure caption

Further issues
- Please remove or explain the abbreviation Tren in the title
- Please use the common abbreviation “SCO” for spin crossover instead of SC, usually used for single crystal
- L 32: Reference needed
- L 54: Remove + after b
- L 55: explain PE
- Fig.1: Please give the IUPAC-names for all shown ligands and draw the complete structure to ease comprehensibility
- L 70: Please specify the search criteria and explain, why those criteria were used
- Table1: Please explain Nap, as this was not done so far. It is not entirely clear, what exactly should be the message of table 1
- L 56-90: See comment on insights-section at the end and remove
- L96-105: Specific examples are needed here
- L106-114: This section is confusing and needs a proper explanation with a graphical support for the reader (fig.1 is not sufficient)
- L114-119: Specific examples are needed here
- Table 2: References missing
- L 187: Explain BM
- L 202-203: See comment on insights-section at the end and remove. Please note: The statement “there is certainly evidence…” and not giving an evidence, is worthless. If there is an evidence for this assumption it should be provided, otherwise the statement is inappropriate
- L 209-210: See comment on insights-section at the end and remove. Please note: If this area may be considered worth further investigation, a clear explanation for this reasoning is necessary
- L219: “A clue that this complex may exhibit…” – Does the Mn-compound exhibit SCO or not? If yes, the evidence is the magnetization curve, no further clue is needed.
- L 235-239: See comment on insights-section at the end and remove
- L 254-256: See comment on insights-section at the end and remove
- L 258-260: This statement was already given several times before – please remove
- L 286-288: I see no reason, why histograms should be considered as clear evidence for two ground states. Please rework this part
- Figure 6-8: A proper explanation what is shown in those figures and especially why this is important is needed.
- Figure 6-8: The labelling of the axes needs to be larger, as is unreadable
- Figure 7: A paragraph detailing the correlation between spin crossover and bite angle is necessary
- Figure 6-8: Please give a detailed interpretation of the figures, not only a description what is shown
- Table 4: Shift to supplement
- L 327-329: The structures and magnetic curves should be shown
- L 334-337: Please use another format for the arrows
- L 339: “A feature to incorporate…” This implies, that a hysteresis can be implemented by design. This is not possible so far, so please rewrite this sentence
- L 361: Please define the abbreviations
- L367-368: As for the mentioned examples references are given, it should be known whether the conformational issues are present or not. Please specify
- L 387-389: In the sentence before the importance of the methyl-group for the spin crossover was explained. The fact, that in one example the methyl-group prevents the spin crossover is contradictive, therefore this result should be discussed in more detail
- L412-415: Why are metallomesogens introduced at this point? Please remove, or give the reader a understanding why this is important at that point
- L427: Please give names or common abbreviations for the solvents. C4H2N2O is not very helpful without searching for the original publication
- L455-458: This is very speculative and should be removed, or detailed further and supported by references
- L614: Change Pbar3
- L647-664: The summary needs a complete rewriting, summarizing the most important aspects of the review article

Author Response

G. Brewer gives a review on structural aspects of tripodal Schiff base complexes featuring spin crossover.
The manuscript is in the present form not suitable for publication. I have several general issues I request to consider, as well as specific ones throughout the manuscript.

All of my responses are colored in blue and any references to line numbers or literature citations are for the current form of manuscript with Track Changes invoked, not the original.

General
- The manuscript is missing a clear structure, starting from the abstract with giving the reader a clear picture what he will learn by reading the review.

The manuscript has been thoroughly revised with an aim to provide a clearer structure.

  • I miss the emphasis on the magnetic aspect of the spin crossover review, as e.g. not a single magnetization curve was represented. As the article was submitted to magnetochemistry, this should be a central aspect. If only structural considerations are the subject of the article, probably a journal as Crystals would be better suited.
  •  
  • Crystals is not a better choice for submission than Magnetochemistry. A response to this point is given  on lines 42-50 and supported by a quote from a Nobel Laureate and the inclusion of three new references (4, 5 and 6) that support the legitimacy of using structural data to examine magnetic properties without inclusion of magnetization curves.   

  • - Please explain all abbreviations throughout the article the first time they are used
  • I believe this has been corrected

  • - The complete article demands for a careful proof-reading. I assume, the author is a native speaker. Nevertheless, there are lots of typos needing corrections, several issues with punctuation (often missing, some superfluous) and phrasing. Please use short sentences and avoid nested sentences, especially when commas are missing. I suggest also using past tense and avoiding the repetitive use of future (e.g. “there will be…”, etc.)
  • The author apologies for the lack of more careful proofreading and believes that this is now corrected.

  • - Of course, it is impossible to mention and describe all examples covered by the topic of a review article. Nevertheless, it is of utmost importance to explain the reader carefully why the selected examples were selected, or why they are considered as important. This step is missing here
  • This point is well taken and modifications have been wade, see for instance, one example line 453-459 and following paragraphs
    - Between all figures and unit symbols (even %) a blank is required
  • These were corrected (perhaps a few were missed)
    - A proof-reading by a native speaker to improve the language-mistakes (tense, plural/singular, phrasing, terminology, …) as this would enhance the readability of the manuscript
  • The author apologies for the lack of more careful proofreading and believes that this is now corrected. Efforts were made or improving the readability'

  • - For all single items / compounds in the tables the references are needed in the tables
  • All the SCO compounds in the Tables are referenced individually in text near the tables where mentioned. Please note that the largest Table (15 examples) now table 3 not 4 was requested to be moved to Supplementary material by this reviewer. See my comment on that point later. If the Table is moved the references need to be in the text for the reader. If the Table is left in the paper, not moved to Supplemental, the references can be pasted into table at that time. This change could be made in Processing stage if manuscript is accepted. Referee II wanted another table deleted so the same reasoning would apply. Leaving tables in text or moving to Supplemental is an Editorial decision and the change to copying references into table (vs in text where discussed) should await that decision. I will do what is needed on that when a decision is made. Very rapid time to change once a decision is reached, keep tables in text or move to Supplement.

  • - All statements (and there are a few in the manuscript) that there would be more work necessary should be removed. A review article is a representation of a status quo, not a personal interpretation of what should or could be done. If there is the urgent need to point out insights from this review, which could be suitable to advance the field, than those should be grouped in a single section at the end with a proper justification.
  • This point is quite valid and the author regrets this mistake and is grateful for the opportunity to correct this at multiple points in the manuscript. All future work/insight statements have been removed from the text of the manuscript. An extremely limited amount was added to the last section and in no more than a few lines, lines 852-862  

  • - All structures should be given as ORTEP-plots (or have a sound explanation why a different representation was chosen), have a short description what exactly is shown and the reader should see, as well have the proper reference in the figure caption
  • ORTEP structures are normally required for the first report of a new structure, which is not the case here. The purpose of the metal complex structures in this review is to illustrate the shape of a molecule, the changes that occur during a SCO (superimposed structures), and the supramoleculat patterns these these SCO molecules that they self assemble into. All illustrated structures of complexes were prepared from the original cif, are perfectly valid representations and were selected to illustrate points that are important. This type of representation of molecules is used in the primary literature for the same reasons it was selected here, it best illustrates the desired point in the discussion. This is given in text on lines 189-195. In this case the use of ORTEPs is inappropriate and would obscure the points you are trying to make.

Further issues
- Please remove or explain the abbreviation Tren in the title

This was removed
- Please use the common abbreviation “SCO” for spin crossover instead of SC, usually used for single crystal

This substitution was done throughout
- L 32: Reference needed

reference 3 line 41 was added
- L 54: Remove + after b the + sign  was removed
- L 55: explain PE This is added lines 33-36
- Fig.1: Please give the IUPAC-names for all shown ligands and draw the complete structure to ease comprehensibility

There are a number of ligands utilized and a supplemental file is added that gives the drawing and name of each which is used throughout the text. The nomenclature issue is explained further in lines 86-98. The use of IUPAC names for these ligands is not commonly (or ever) done in the primary literature. An example is given on lines  97 and 98 to support this. Simple abbreviations and the supplemental file that gives the structure and name of each is the best way to go and is commonly done in this area. This will increase comprehensibility much more than the use of IUPAC names of these ligands. IUPAC ligand names are not used in the primary literature for this class of ligand outside of a pure structure report. 
- L 70: Please specify the search criteria and explain, why those criteria were used Please see lines 109-117
- Table1: Please explain Nap, as this was not done so far. It is not entirely clear, what exactly should be the message of table 1 Nap is now defined in the caption of Table 1. The purpose of the table is to give the reader a quick picture of the number and structural variations that exist for this type of tridodal ligand class.  
- L 56-90: See comment on insights-section at the end and remove

  • This point is quite valid and the author regrets this mistake and is grateful for the opportunity to correct this at multiple points in the manuscript. All future work/insight statements have been removed from the text of the manuscript. An extremely limited amount was added to the last section and in no more than a few lines, lines 852-862  

  • L96-105: Specific examples are needed here
  • It is hoped that the Supplemetal file with names will address this concern as will the other many examples of ligands throughout the manuscript should. 
    - L106-114: This section is confusing and needs a proper explanation with a graphical support for the reader (fig.1 is not sufficient)
  • There is now a supplemental file for ligands employed and please see lines 86-98

  • - L114-119: Specific examples are needed here
  • Examples are provided in Figures 2 and 3.
    - Table 2: References missing
  • Cross references now in capton
    - L 187: Explain BM given in line 252-3
    - L 202-203: See comment on insights-section at the end and remove. Please note: The statement “there is certainly evidence…” and not giving an evidence, is worthless. If there is an evidence for this assumption it should be provided, otherwise the statement is inappropriate
  • This point is quite valid and the author regrets this mistake and is grateful for the opportunity to correct this at multiple points in the manuscript. All future work/insight statements have been removed from the text of the manuscript. An extremely limited amount was added to the last section and in no more than a few lines, lines 852-862  

  • - L 209-210: See comment on insights-section at the end and remove. Please note: If this area may be considered worth further investigation, a clear explanation for this reasoning is necessary
  • This point is quite valid and the author regrets this mistake and is grateful for the opportunity to correct this at multiple points in the manuscript. All future work/insight statements have been removed from the text of the manuscript. An extremely limited amount was added to the last section and in no more than a few lines, lines 852-862  

  • - L219: “A clue that this complex may exhibit…” – Does the Mn-compound exhibit SCO or not? If yes, the evidence is the magnetization curve, no further clue is needed.
  • This point is quite valid and the author regrets this mistake and is grateful for the opportunity to correct this at multiple points in the manuscript. All future work/insight statements have been removed from the text of the manuscript. An extremely limited amount was added to the last section and in no more than a few lines, lines 852-862. The cpd in question is HS and this is stated on lines 341-343  

  • - L 235-239: See comment on insights-section at the end and remove
  • This point is quite valid and the author regrets this mistake and is grateful for the opportunity to correct this at multiple points in the manuscript. All future work/insight statements have been removed from the text of the manuscript. An extremely limited amount was added to the last section and in no more than a few lines, lines 852-862  

  • - L 254-256: See comment on insights-section at the end and remove
  • This point is quite valid and the author regrets this mistake and is grateful for the opportunity to correct this at multiple points in the manuscript. All future work/insight statements have been removed from the text of the manuscript. An extremely limited amount was added to the last section and in no more than a few lines, lines 852-862  

  • - L 258-260: This statement was already given several times before – please remove Statement removed
    - L 286-288: I see no reason, why histograms should be considered as clear evidence for two ground states. Please rework this part
  • This is an extremely important point and this has been expanded. Please see lines 368-405. Figures 6, 7 and 8 have all been redone at higher quality. A bimodal max in a histogram plot of a structural parameter of the atom attached to an atom undergoing a spin state change is evidence of two different ground states as discussed in lines 368-405. A bond length or angle is a measurement of the ground state. If there are two Max in a tightly defined group of compounds then there are two ground states. If there are 2 ground states then a SCO is possible. 
    - Figure 6-8: A proper explanation what is shown in those figures and especially why this is important is needed.
  • This is an extremely important point and this has been expanded. Please see lines 368-405. Figures 6, 7 and 8 have all been redone at higher quality. A bimodal max in a histogram plot of a structural parameter of the atom attached to an atom undergoing a spin state change is evidence of two different ground states as discussed in lines 368-405. A bond length or angle is a measurement of the ground state. If there are two Max in a tightly defined group there are two ground states. If there are 2 ground states then a SCO is possible. 
    - Figure 6-8: The labelling of the axes needs to be larger, as is unreadable
  • This is an extremely important point and this has been expanded. Please see lines 368-405. Figures 6, 7 and 8 have all been redone at higher quality. A bimodal max in a histogram plot of a structural parameter of the atom attached to an atom undergoing a spin state change is evidence of two different ground states as discussed in lines 368-405. A bond length or angle is a measurement of the ground state. If there are two Max in a tightly defined group there are two ground states. If there are 2 ground states then a SCO is possible. 
    - Figure 7: A paragraph detailing the correlation between spin crossover and bite angle is necessary
  • Please see lines 401-405. The correlation is between bite angle and spin state, not between bite angle and SCO. In this class of ligand if a molecule exhibits two different bite  angles  at two different temperatures then there are 2 ground states A SCO exists in between these 2 temperatures.  
  • This is an extremely important point and this has been expanded. Please see lines 368-405. Figures 6, 7 and 8 have all been redone at higher quality. A bimodal max in a histogram plot of a structural parameter of the atom attached to an atom undergoing a spin state change is evidence of two different ground states as discussed in lines 368-405. A bond length or angle is a measurement of the ground state. If there are two Max in a tightly defined group there are two ground states. If there are 2 ground states then a SCO is possible. 
    - Figure 6-8: Please give a detailed interpretation of the figures, not only a description what is shown
  • This is an extremely important point and this has been expanded. Please see lines 368-405. Figures 6, 7 and 8 have all been redone at higher quality. A bimodal max in a histogram plot of a structural parameter of the atom attached to an atom undergoing a spin state change is evidence of two different ground states as discussed in lines 368-405. A bond length or angle is a measurement of the ground state. If there are two Max in a tightly defined group there are two ground states. If there are 2 ground states then a SCO is possible. 
    - Table 4: Shift to supplement
  • This is now Table 3 not 4. Please see comment on this given earlier as there is overrlap between points. All the SCO compounds in the Tables are referenced individually in text near the tables where mentioned. Please note that the largest Table (15 examples) now table 3 not 4 was requested to be moved to Supplementary material by this reviewer. See my comment on that point later. If the Table is moved the references need to be in the text for the reader. If the Table is left in the paper, not moved to Supplemental, the references can be pasted into table at that time. This change could be made in Processing stage if manuscript is accepted. Referee II wanted another table deleted so the same reasoning would apply. Leaving tables in text or moving to Supplemental is an Editorial decision and the change to copying references into table (vs in text where discusses) should await that decision. I will do what is needed on that when a decision is made. Very rapid time to change once the above is decided.

  • - L 327-329: The structures and magnetic curves should be shown
  • The structure is provided in Supplement and the magnetic properties described, please see lines 455-458
    - L 334-337: Please use another format for the arrows These are equilibrium arrows from WORD that I have available
    - L 339: “A feature to incorporate…” This implies, that a hysteresis can be implemented by design. This is not possible so far, so please rewrite this sentence Please see lines 476-480, the mistake is removed
    - L 361: Please define the abbreviations All abbreviations are defined in text when used, please see lines 490-506.
    - L367-368: As for the mentioned examples references are given, it should be known whether the conformational issues are present or not. Please specify. There are conformational issues invlolved and this is why the relaxation time is long in one case. This is correct and this commented on in lines 502-506
    - L 387-389: In the sentence before the importance of the methyl-group for the spin crossover was explained. The fact, that in one example the methyl-group prevents the spin crossover is contradictive, therefore this result should be discussed in more detail
  • The author is grateful for catching this point. Please see lines 545-554. You are correct that the 2nd MePy complex is SCO but does not become LS until very low T whereas other examples are LS at 100-150 K.
    - L412-415: Why are metallomesogens introduced at this point? Please remove, or give the reader a understanding why this is important at that point. The 1st use of term, metallomesogens, gas now been moved earlier to line 512
    - L427: Please give names or common abbreviations for the solvents. C4H2N2O is not very helpful without searching for the original publication
  • Thanks for catching this. The formula of  solvate is C4H4N2O, imidazole-4-carboxaldehyde. This is corrected, please see lines 587-589 and lines 594-597. 
    - L455-458: This is very speculative and should be removed, or detailed further and supported by references
  • This is not speculative at all. It is based on two structures discussed in the section and there are two references added that discuss this type of change (environmental) causing a change in spin state. Please see lines 616-625 and 2 references in line 625 
    - L614: Change Pbar3 corrected/removed, not needed
    - L647-664: The summary needs a complete rewriting, summarizing the most important aspects of the review article
  • This was completely redone, please see lines 818-862

Reviewer 2 Report

This manuscript reviews the recent chemistry and structural parameters affecting the spin state of tripodal Schiff base complexes of tren and related amines. The article covers mainly the first row transition metal complexes and tries to clarify the low spin - high spin relationship in these compounds by examining the structural factors. I have few concerns before publication listed below:

The introduction gives a brief overview of SC complexes but I would have liked a little bit more justification as to why it is an important field of study. In the line 56 the author introduces donor sets but I don't understand this naming convention and it is not clarified anywhere? Also, the author uses many different abbreviations throughout the text but doesn't elaborate what they mean. The figures in the manuscript need to be labelled better, for example in Figure 1, better naming system is needed, the reader is very confused by the current labeling. All of the figures and tables are missing references. Figure 2 (and similar later) needs colour coded labels, lists of parameters is not helpful, when the reader does not know which atoms are which. In Figure 5, could the difference just be due to different measuring temperatures? Table 3 has a lot of data averaged, I don't think there is any additional information gained by averaging all of the 17 compounds and parameters to one value? Line 252: which deviations is the author talking about? Line 262: I don't understand the sentence beginning "The hemiprotonated..." ChemDraw pictures of the complexes rather than names would be better. The author lists parameters in the figure captions but doesn't discuss about them in the text, I thought this was the whole point of the manuscript? The author gives a lot of attention to supramolecular/dinuclear complexes but I would have liked a bit more focus on how they determined that these species are SC complexes.

Author Response

This manuscript reviews the recent chemistry and structural parameters affecting the spin state of tripodal Schiff base complexes of tren and related amines. The article covers mainly the first row transition metal complexes and tries to clarify the low spin - high spin relationship in these compounds by examining the structural factors. I have few concerns before publication listed below:

My responses are in blue and line numbers and citation numbers are to the revised form, with Track Changes activated, not the original submission.

The introduction gives a brief overview of SC complexes but I would have liked a little bit more justification as to why it is an important field of study.

This is given in references 9, 10, 11, 12 and 13 which deal largely with electronic and materials science applications. The concept is that a single SCO molecule can serve as a stitch or storage device of one bit of information. 

In the line 56 the author introduces donor sets but I don't understand this naming convention and it is not clarified anywhere?

This is now corrected. This is given in lines 75-81

Also, the author uses many different abbreviations throughout the text but doesn't elaborate what they mean.

Corrected. All abbreviations are now defined the first time they are used.

The figures in the manuscript need to be labelled better, for example in Figure 1, better naming system is needed, the reader is very confused by the current labeling. All of the figures and tables are missing references. Figure 2 (and similar later) needs colour coded labels, lists of parameters is not helpful, when the reader does not know which atoms are which.

The nomenclature of ligands is expanded, lines 86-98. In addition a large table giving a drawing of each ligand ant its name is now provided as a supplement. Figures 2 and 3 have references and the structural parameters are discussed more in text. Hopefully this  well help. The only atoms bound to the metal (donor atoms) are N (in blue) in figures 2 and 3. 

In Figure 5, could the difference just be due to different measuring temperatures?

Yes the difference is due entirely to temperature. At the lower temp you have the LS as the ground state  and at the higher temp the HS as the ground state. The difference in bond lengths is ~ 0.2 Angstroms for Fe(II) or Fe(III) in going from LS to HS. For Mn(III) in Fig 4 the difference is smaller ~0.1 A as this is a one vs a 2 electron change.

Table 3 has a lot of data averaged, I don't think there is any additional information gained by averaging all of the 17 compounds and parameters to one value?

Thanks for this and the Table has been removed

Line 252: which deviations is the author talking about?

This portion of text was deleted as it referred to Table 3 (please see above) which you believed provided little value to the manuscript. On reflection Table 3 and subsequent comments were not needed.

Line 262: I don't understand the sentence beginning "The hemiprotonated..." ChemDraw pictures of the complexes rather than names would be better.

A large supplemented has been added and it giver structural diagrams of the ligands utililized including the hemidepronated ones you mention. Names for each are provided that are used throughout the text.

You can think of hemideprotonated as a 50:50 mixture of H3L and L3-, which averages to H1.5L 1.5-

The author lists parameters in the figure captions but doesn't discuss about them in the text, I thought this was the whole point of the manuscript?

This has been corrected for several of the figures that provide data for the LS and HS form of the same molecule.  The data is now in text below figure and values discussed in terms of agreement with the theoretical values from Table 2.

The author gives a lot of attention to supramolecular/dinuclear complexes but I would have liked a bit more focus on how they determined that these species are SC complexes.

This is mentioned in lines 731-737. Magnetic susceptibility was used as well but was less distinctive as the SCO were often gradual (not sharp) and in some case there were different transitions that overlapped. The main point in the supramoleclar section was how the topology of the large assembly depended on the building block used.

Round 2

Reviewer 1 Report

I would be happy to suggest it now for publication. The author went through all my issued points carefully, enhancing the quality of the manuscript.

Reviewer 2 Report

I am satisfied with the changes made.